# Fuzhengjiedu San inhibits porcine reproductive and respiratory syndrome virus by activating the PI3K/AKT pathway

**Kexin Chang**[1], **Kuangshi Fan**[1], **Hua Zhang**[1], **Qiong Wu**[1], **Yonghong Zhang**[1], **Le Wang**[1], **Hongcen Chen**[1], **Jinjin Tong**[2]*, **Defeng Cui**[1]*

1 Beijing Key Laboratory of Traditional Chinese Veterinary Medicine, Animal Science and Technology College, Beijing University of Agriculture, Beijing, PR China, 2 Beijing Key Laboratory of Dairy Cow Nutrition, Animal Science and Technology College, Beijing University of Agriculture, Beijing, PR China

☯ These authors contributed equally to this work.
* cdfffff@163.com (DC); tongjinjin0451@163.com (JT)

## Abstract

### Background

Traditional Chinese medicine (TCM) has been garnering ever-increasing worldwide attention as the herbal extracts and formulas prove to have potency against disease. Fuzhengjiedu San (FZJDS), has been extensively used to treat viral diseases in pigs, but its bioactive components and therapeutic mechanisms remain unclear.

### Methods

In this study, we conducted an integrative approach of network pharmacology and experimental study to elucidate the mechanisms underlying FZJDS's action in treating porcine reproductive and respiratory syndrome virus (PRRSV). We constructed PPI network and screened the core targets according to their degree of value. GO and KEGG enrichment analyses were also carried out to identify relevant pathways. Lastly, qRT-PCR, flow cytometry and western blotting were used to determine the effects of FZJDS on core gene expression in PRRSV-infected monkey kidney (MARC-145) cells to further expand the results of network pharmacological analysis.

### Results

Network pharmacology data revealed that quercetin, kaempferol, and luteolin were the main active compounds of FZJDS. The phosphatidylinositol-3-kinase (PI3K)/Akt pathway was deemed the cellular target as it has been shown to participate most in PRRSV replication and other PRRSV-related functions. Analysis by qRT-PCR and western blotting demonstrated that FZJDS significantly reduced the expression of P65, JNK, TLR4, N protein, Bax and IBa in MARC-145 cells, and increased the expression of Bcl-2, consistent with network pharmacology results. This study provides that FZJDS has significant antiviral activity through its effects on the PI3K/AKT signaling pathway.

**Data Availability Statement:** All relevant data are within the paper and its Supporting Information files.

**Funding:** This study was financially supported by the Swine Innovation Team Project of the Beijing Modern Agricultural Industrial Technology System (BAIC02) and the Swine Innovation Team Project of Beijing (BAIC05-2022).

**Competing interests:** The authors have declared that no competing interests exist.

**Abbreviations:** TCM, traditional Chinese medicine; FZJDS, Fu Zheng Jie Du powder; PRRSV, porcine reproductive and respiratory syndrome virus; MLV, modified live virus; TCM, traditional Chinese medicine; APS, astragalus polysaccharide; RIPs, radix isatidis polysaccharides; PCV2, porcine circovirus 2; MDA, malondialdehyde; ROS, reactive oxygen species; GSH, glomerular-stimulating hormone; SOD, superoxide dismutase; HSV, herpes simplex virus; IC50, half maximal inhibitory concentration of a substance; NKs, natural killer cells; IFN, interferon; TNF, tumor necrosis factor; IL-2, interleukin-2; COVID-19, corona virus disease-2019; TCMSP, traditional Chinese medicine systems pharmacology database and analysis platform; ETCM, encyclopedia of traditional Chinese medicine; QED, quantitative estimate of drug-likeness; HERB, A high-throughput experiment- and reference-guided database of traditional Chinese medicine; OB, oral bioavailability; DL, drug similarity; CTD, comparative toxicogenomics database; OMIM, online Mendelian inheritance in man; PPI, protein-protein interaction; TSV, tab-separated values; GO, gene ontology; KEGG, Kyoto Encyclopedia of Genes and Genomes; DAVID, Database for Annotation, Visualization and Integrated Discovery; STRING, Search Tool for the Retrieval of Interacting Genes/Proteins; FDR, false discovery rate; DMEM, Dulbecco's modified Eagle's medium; FBS, fetal bovine serum; DMSO, dimethyl sulfoxide; JNK, c-Jun N-terminal kinase; IgG, immunoglobulin G; qRT-PCR, quantitative real-time polymerase chain reaction; CPE, cytopathic effect; TCID50, median tissue culture infective dose; CCK-8, cell counting kit-8; PMSF, phenylmethylsulfonyl fluoride; SDS-PAGE, sodium dodecyl sulfate-polyacrylamide gel electrophoresis; PVDF, polyvinyl difluoride; BSA, bovine serum albumin; TBST, Tris-buffered saline with Tween 20; ECL, enhanced chemiluminescence; ANOVA, analysis of variance; ACTB, β-Actin; TP53, tumor protein p53; CASP3, caspase-3; MAPK, mitogen-activated protein kinase; IL-6, interleukin-6; MYC, Myc proto-oncogene protein; RELA, NF-kappaB transcription factor p65 subunit; GSK3B, glycogen synthase kinase3β; PPARG, peroxisome proliferative activated receptor, gamma; CASP8, caspase-8; OD, optical density; SD, standard

## Conclusion

We conclude that FZJDS is a promising candidate herbal formulation for treating PRRSV and deserves further investigation.

## 1. Introduction

Porcine reproductive and respiratory syndrome virus (PRRSV) can replicate and spread throughout the body, causing organ damage, inflammation, immunosuppression, persistent infection and secondary infections. Viral infection constitutes a serious threat to the health of swine and can result in huge economic losses to the industry. PRRSV is mainly prevented by vaccine at present. However, facing up immunosuppression caused by PRRSV infection, virus evolution and multiple recombination between wild-type and modified live virus (MLV), it can not be effectively prevented.

Traditional Chinese medicine (TCM) offers a huge pharmacopoeia of compounds and formulations rich in active ingredients and pharmacological effects, with little toxicity to animals. Fu Zheng Jie Du San (FZJDS) is a classic TCM herbal formula, which is composed of *Hedysarum multijugum* Maxim, *Isatidis radix* and *Epimedii herb*a. FZJDS has been recognized as having immunomodulatory activity, and is able to restore immune cell function and induce the body to produce interferons with potent therapeutic antiviral activity against PCV2 and other viral diseases. The ingredients in FZJDS have been widely used in antiviral formulations with astragalus polysaccharide (APS) and significantly decreased PCV2 DNA replication, the level of *MDA* and *ROS*, and the activation of *NF-κB* in porcine kidney-15 cells. APS significantly increased *GSH* and *SOD* which inhibited PCV2 infection [1]. Previous studies reported that APS inhibited the proliferation of astrocytes infected by the herpes simplex virus [2]. The active compound of *radix isatidis* (BanLanGen), 3-(furan-2-yl)-7-hydroxyisoquinolin-1(2H)-one, has shown potent anti-HSV-1 activity with an IC50 value of 15.3 µg/mL [3]. *Radix isatidis* polysaccharides (RIPs) can promote the proliferation of NK cells and the expression of *IFN-γ* and *TNF-α*, to regulate the immune system by activating dendritic cells, macrophages and lymphocytes [4].

Epimedin C is a major flavonoid glycoside derived from *Herba epimedii*, which is able to modulate the immune response by increasing lymphocyte proliferation and production of *IL-2* in a murine model [5]. However, previous reports have only described the anti-PRRSV activity of HuangQi, BanLanGen and YinYangHuo separately; it remains to be confirmed whether the formulation of FZJDS has potent antiviral affects against PRRSV.

Network pharmacology is a new discipline that integrates transcriptomics, proteomics, and metabolomics to provide a powerful tool for researchers to determine the therapeutic mechanism and clarify the multiple components and targets of specific compounds from TCM. Using the tool of network pharmacology, it has been reported that the LianHuaQingWen herbal compound possesses the ability to modulate the inflammatory response, has proven anti-COVID-19 effects, and the capability to repair lung injury [6]. Network pharmacology analysis also found that the ShuangHuangLian oral liquid had therapeutic anti-inflammatory properties and antiviral activity against COVID-19 [7]. In another study, the network pharmacology technique along with experimental validation provided evidence that the Qingfei oral liquid (QF) was effective in the treatment of respiratory syncytial virus-induced lung inflammation through its action on the PI3K/Akt/mTOR signaling pathway [8]. Active ingredients based on total flavones were shown to provide profound protection against pulmonary

deviation; BP, biological processes; CC, cellular components; MF, molecular functions; TLR, Toll-like receptor; AR, aldose reductase; JUN, jun proto-oncogene; COX-2, cyclooxygenase-2; F3, coagulation factor III; CYP1A1, cytochrome p450 family 1 subfamily A polypeptide 1; TGFB1, transforming growth factor beta-1; IFNG, interferon gamma; CCND1, cyclin D1; HIF1A, hypoxia-inducible factor 1 alpha; CASP-9, caspase-9; IL-10, interleukin-10; HMOX1, Heme oxygenase (decycling) 1; FOS, proto-oncogene c-Fos; BCL2, B cell lymphoma/ leukemia2; BAX, BCL2-associated X protein; NOX2, nicotinamide adenine dinucleotide phosphate oxidase 2; CDK1, cyclin-dependent kinase-1.

influenza A virus (IAV) infection by alleviating inflammatory responses, decreasing signaling through the MAPK pathway and expediting viral eradication [9]. The distinct superiority of network pharmacological analysis ensures that we can directly achieve experimental verification based on the results returned by the network analysis. Thus, a network pharmacologic approach was used to predict the FZJDS targets and gene interactions necessary for effective therapy against PRRSV. Our research provides additional insights into the physiological action of FZJDS, which should stimulate improvements in the formulation for better anti-PRRSV outcomes, and will serve as a useful reference for exploring the therapeutic mechanisms of other traditional Chinese medicines.

## 2. Materials and methods

### 2.1. Screening FZJDS for active compounds against putative targets in PRRVS

In this study, the traditional Chinese medicine systems pharmacology database, **TCMSP**, (https://old.tcmsp-e.com/tcmsp.php) [10], the encyclopedia of traditional Chinese medicine, **ETCM** (http://www.tcmip.cn/ETCM/index.php/Home/), and a high-throughput experiment- and reference-guided database of traditional Chinese medicine, **HERB** (http://herb.ac.cn/) were consulted to identify the chemical components of FZJDS. The parameters of oral bio-availability (OB) >30% and drug likeness (DL) >0.18 were set up as the criteria in TCMSP to screen for active ingredients. The ingredients collected in the ETCM database were divided into three categories according to a quantitative estimate of drug likeness (QED): weak (QED<0.49), moderate (0.49≤QED≤0.67) and good (QED>0.67). We limited the search to a drug likeness of moderate to good. The HERB database was used to supplement the first two, and a list of potential targets of the identified components was obtained from these three databases after deleting repetitive targets.

### 2.2. Potential PRRVS targets of FZJDS

All specific PRRVS targets were obtained from a search of the human genetic database, **Gene-Cards** (https://www.genecards.org/), the comparative toxicogenomics database, **CTD** (http://ctdbase.org/), and the online Mendelian inheritance in man platform, **OMIM** (http://omim.org/) [11]. These databases illuminate the relationship between targets and diseases from different perspectives. GeneCards is a free, authoritative, comprehensive database which provides genomics, transcriptomics, proteomics, etc. CTD contains accurate data and descriptions of chemical/gene/protein interactions and relationships between specific chemicals and diseases across species, as well as genetic associations with diseases. OMIM focuses on the association between disease phenotypes and the genes linked to disease etiology and progression. Potential targets were subjected to further analysis after removing repetitive items [12].

### 2.3. Construction of protein-protein interaction network and screening of core targets

The general targets of the active ingredients of FZJDS and the specific PRRSV-related targets were compared to obtain a Venn diagram of common potential targets for FZJDS treatment of PRRS. The FZJDS active ingredient targets and PRRSV-specific targets were imported into VENNY2.1 (https://bioinfogp.cnb.csic.es/tools/venny/) for visualization. Then, the STRING database (https://string-db.org/) [13] was used to construct a protein-protein interaction (PPI) network by overlapping the drug-related targets with the PRRSV targets. Species were limited to *Sus scrofa* with a minimum required interaction score >0.4. Other parameters in the study

were included at their default setting. The TSV formatted file was downloaded from the STRING database and imported into Cytoscape 3.7.2 (http://www.cytoscape.org/) [14], which is an open-source software platform that is used for visualizing complicated biomolecular networks and integrating different types of attribute data. A build-in module, CytoCNA, was utilized to generate major network targets when the "between-ness", closeness and degree of each node were all larger than the median. Core target networks were generated from major target networks by topological analysis.

## 2.4. Constructing the interaction network between drug-related and PRRVS- related targets

A depiction of the drug-PRRSV common targets network was created using the data obtained from the aforementioned datasets and the Cystoscape 3.7.2 software package to visualize the complex relationships among drugs, PRRSV and their related targets. In this network, nodes represent the drugs, PRRSV and their related targets, while the connections between the nodes represent the known interactions between them.

## 2.5. Gene ontology (GO) and Kyoto Encyclopedia of Genes and Genomes (KEGG) enrichment analysis of targets common to both FZJDS and PRRSV

The functional annotation tool, DAVID (https://david.ncifcrf.gov/) [15], was utilized to perform GO and KEGG pathway analysis, the organism was set as *Sus scrofa*, significance at $P<$ 0.01 and FDR<0.05 for differential screening in GO analysis, and depicting the top ten collected terms in descending order in a bar chart. Similarly, significance was set at $P<0.01$ for KEGG analysis after removing human disease-related pathways. The top twenty pathways selected in ascending order of $P$ were visualized as a bubble chart with an online bioinformatics tool (http://www.bioinformatics.com.cn/).

## 2.6. Construction of an FZJDS-PRRSV-targets-pathways network

The top twenty enriched KEGG pathways from enrichment analysis corresponding to $P$, together with the identified core targets were loaded into Cytoscape 3.7.2, and by separately adjusting the parameters, a visualized plot of the *FZJDS* -PRRSV-targets-pathways network was constructed.

## 2.7. In vitro cell culture and PRRSV propagation

MARC-145 cells, derived from African green monkey kidney, were cultivated in Dulbecco's modified Eagle's medium (DMEM, Gibco) supplemented with 10% fetal bovine serum (FBS, Gibco) and penicillin-streptomycin (Gibco) at 37˚C in a humidified 5% $CO_2$ atmosphere. PRRSV strain JXA-1 (GenBank accession no. EF112445.1; lineage 8.7) was propagated in MARC-145 cells, and virus titers were calculated using the Reed-Muench method.

## 2.8. Reagents and antibodies

The NF-B inhibitor, BAY 11–7082 (Cat. No. S1523), and the JNK inhibitor, SP600125 (Cat. No. S1876), were purchased from Beyotime (China). All drugs were prepared in DMSO according to the company's instruction. Anti-PRRSV-N rabbit polyclonal antibody (bs-23941R) and anti-TLR4 rabbit monoclonal antibody (bs-20379R) were purchased from Bioss (China). Anti-Bax mouse monoclonal IgG antibody (sc-7480), anti-JNK1/2(D-9) mouse monoclonal IgG antibody (sc-137019), anti-phospho-JNK (G-7) mouse monoclonal IgG

antibody (sc-6254) and anti-Bcl-2 mouse monoclonal IgG antibody (sc-7382) were purchased from Santa Cruz (Santa Cruz, USA). Anti-β-Actin rabbit monoclonal antibody (AF5003), anti-phospho-NF-B p65 rabbit monoclonal antibody (Ser536), anti-NF-kB p65 rabbit monoclonal antibody (AF1234), goat anti-rabbit IgG antibody (A0208) and goat anti-mouse IgG antibody(A0216) were all purchased from Beyotime (China). Anti-phospho-IBa (S32) rabbit mAb(14D4) and anti-IBa mouse mAb(L35A5) were purchased from Cell Signaling Technology (CST, USA).

## 2.9. Quantitative real-time PCR (qRT-PCR)

Total virus RNA was extracted using a total RNA rapid extraction kit (Accurate Biology, HuNan, China) according to the instructions, and 1 μg RNA of each sample was subsequently reverse transcribed to cDNA with a reverse transcription kit (Accurate Biology, HuNan, China) according to instructions. Further detection was performed according to the qRT-PCR for PRRSV viral load by established assays [16].

## 2.10. PRRSV infection and virus titration

Cells were infected with type 2 PRRSV and cultured for up to 24 h in Opti-MEM (Gibco). Plaque assays were used to determine virus titer, according to standard methods. In brief, the concentration of Marc-145 cell grafting was adjusted to $1 \times 10^5$/ml and cultured in flat 96-well cell culture plate. Confluent monolayers of MARC-145 cells were inoculated with 200 μl of each sample and incubated for 1 h under the same culture conditions as described above. After incubation, the inoculum was discarded and replaced with DMEM, the plates were incubated for an additional three days, and monitored for cytopathic effects (CPE) daily. The Reed–Muench method was used to calculate the 50% tissue culture infected dose (TCID50). The titer of each virus sample was calculated based on CPE and was expressed as 50% tissue culture infective dose ($TCID_{50}$)/ml. GraphPad Prism 8.0.2 software was used to estimate the 50% inhibition concentration (IC50) of each compound.

## 2.11. Cytotoxicity assay

The cytotoxicity of FZJDS was measured by using the CCK-8 cell proliferation and cytotoxicity assay kit (Beyotime, China). MARC-145 cells were seeded in 96-well plates ($2 \times 10^4$ cells per well) and grown at 37˚C. Cells were incubated with medium supplemented with different concentrations of FZJDS for 48 h. The absorbance of each well was read at 570 nm with a reference wavelength of 630 nm using a microplate reader. The 50% cytotoxic concentration ($CC_{50}$) was calculated by the GraphPad Prism 8.0.2 software. The selectivity index (SI) was determined by the ratio of CC50 to IC50.

## 2.12. Protein extraction and western blotting

Six-well plate cell samples ($2 \times 10^6$ cells/well) were lysed in cell lysis buffer (Beyotime, China) containing phenylmethylsulfonyl fluoride (PMSF) (Beyotime, China) and phosphatase inhibitors. Cell lysates were subjected to 10–15% sodium dodecyl sulfate-polyacrylamide gel electrophoresis (SDS-PAGE) and transferred by semi-dry blotting onto a polyvinyl difluoride (PVDF) membrane (Merck Millipore, USA). Membranes were blocked with 3% bovine serum albumin (BSA) (Beyotime, China) in TBST (20 mM Tris-HCl, pH 8.0, 150 mM NaCl, 0.05% Tween 20) for 1 h at room temperature. After blocking, membranes were probed with the indicated primary antibodies overnight at 4˚C. After thorough washing, the membranes were incubated with the corresponding secondary antibodies for 1 h at 37˚C. The protein bands

were visualized using an enhanced chemiluminescence (ECL) reagent (Vazyme, China), and a chemiluminescence imaging system (0I-X6, China) was used to analyze the PVDF membranes. Densitometry of protein band intensity was performed using ImageJ software from NCBI.

### 2.13. Flow cytometry

MARC-145 cells were harvested and fixed with paraformaldehyde for 10 min at RT, gently rinsed three times with PBS, and stained with antibodies in 1% BSA for 1 h at room temperature. After rinsing three times with PBS, cells were resuspended in PBS at a concentration of $1\times10^6$ cells/mL. Data were collected on a FACSCalibur flow cytometer (BD LSRII, USA), and analyses were performed using the FlowJo software.

### 2.14. Statistical analysis

Data from each experiment was subjected to one-way analysis of variance (ANOVA) followed by Tukey's $t$ test using Graphpad Prism 8.0 software. $P < 0.05$ was considered to be significant. Significant differences compared with the control group are denoted by* $P < 0.05$, ** $P < 0.01$, *** $P < 0.001$ and **** $P < 0.0001$.

## 3. Results

### 3.1. Screening the active compounds and potential targets

The FZJDS formulation consists of HuangQi (*Hedysarum multijugum Maxim.*), BanLanGen (*Isatis indigotica*) and YinYangHuo (*Epimedii herba*) (**Table 1**), the multiple components and various targets of FZJDS against PRRSV were identified. A total of 82 compounds were collected from the TCMSP, HERB and ETCM databases. Out of these 82 compounds, 20 were from HuangQi, 39 from BanLanGen, and 23 were from YinYangHuo. After removing duplications and combining, the search results were collected from the OMIM, CTD and GeneCards databases. A total of 880 corresponding potential therapeutic targets were found and transformed to gene symbols using the STRING database. By performing an intersection comparison between the drug targets and the disease-associated targets, 30 common core genes shared between FZJDS and PRRSV were identified (**Fig 1**).

### 3.2. Protein-protein interaction network and components-disease network

The protein-protein interaction (PPI) network constructed using the STRING database showed that the 30 intersecting core proteins encoded by these target genes had complex interactions (**Fig 2A**). As obtained using the String database, the PPI network has 30 nodes and 249 edges; the average node degree of the constructed network was 16.6. The PPI network was imported into Cytoscape to calculate the topological parameters for further analysis, and a key subnetwork composed of twelve target genes was obtained by using CytoNCA (**Fig 2B, Table 2**).

**Table 1. Composition of FZJDS.**

| Chinese name | Genus/species | English name | Medicinal part |
|:---:|:---:|:---:|:---:|
| Huang Qi | *Hedysarum multijugum Maxim* | Sweetvetch | Root |
| Ban Lan Gen | *Isatis indigotica* | Chinese woad | Root |
| Yin Yang Huo | *Epimedii herba* | Epimedium | Root |

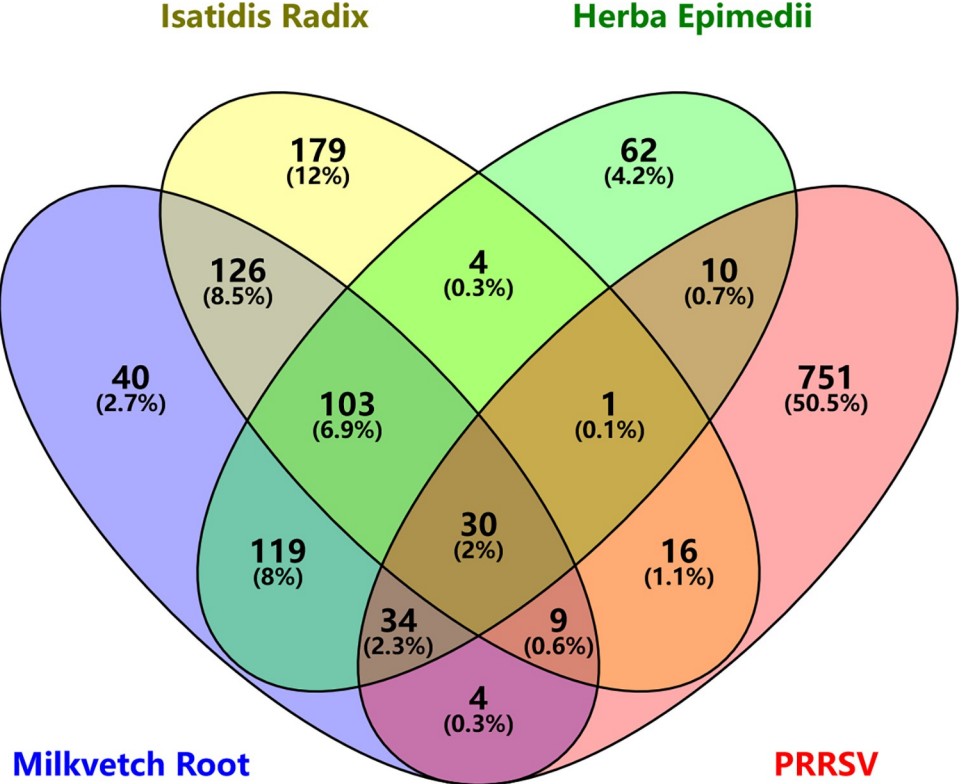

**Fig 1. Venn diagram showing intersection of common targets of FuZhengJieDu (drug) and PRRSV (disease).**

In the present study, Cytoscape 3.7.2 was used to construct a network of the 30 core targets in common of the active ingredients in FZJDS and the PRRSV. The resulting network had 83 nodes and 312 edges, and the connections between the nodes indicated the relationships among the active ingredients and viral targets (**Fig 3**). The more connections, the more important the nodes are in the network. From the network, 52 active compounds with 30 corresponding hub targets were identified. More importantly, by further analyzing the degrees of each node, we found that quercetin, kaempferol and luteolin were likely the key active compounds of FZJDS. Interestingly, the core targets as determined in the same manner have revealed that *COX-2*, *NOS2*, *CDK1* and *AR* were also key targets for PRRSV treatment (**Fig 3**, **Table 3**).

### 3.3. GO enrichment and KEGG pathway analysis

The 30 core intersecting genes were submitted for GO and KEGG enrichment analysis by using DAVID. GO analysis yielded entries comprising 122 biological processes (BPs), 7 cellular components (CCs) and 21 molecular functions (MFs). Different categories of BP, CC, and MF are represented by green, orange and purple bars, respectively. The height of the bar represents the enrichment score observed in the category. By setting the filter for $P<0.05$, the top 10 terms significantly enriched in BPs and MFs were obtained according to fold-enrichment from high to low. All CC categories are displayed visually, and 25 significantly enriched GO terms were obtained (**Fig 4A**). A total of 101 terms were obtained from KEGG pathway enrichment analysis by using DAVID data ($P < 0.01$, FDR $< 0.05$). The results showed that these target genes were in the pathways related to viral infection, the differentiation of immune cells

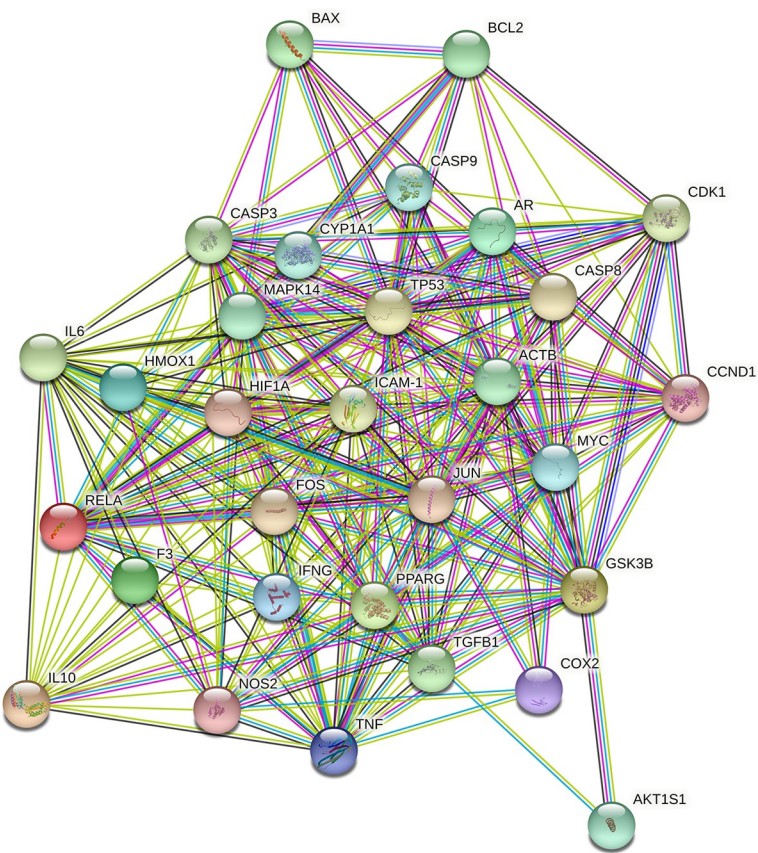

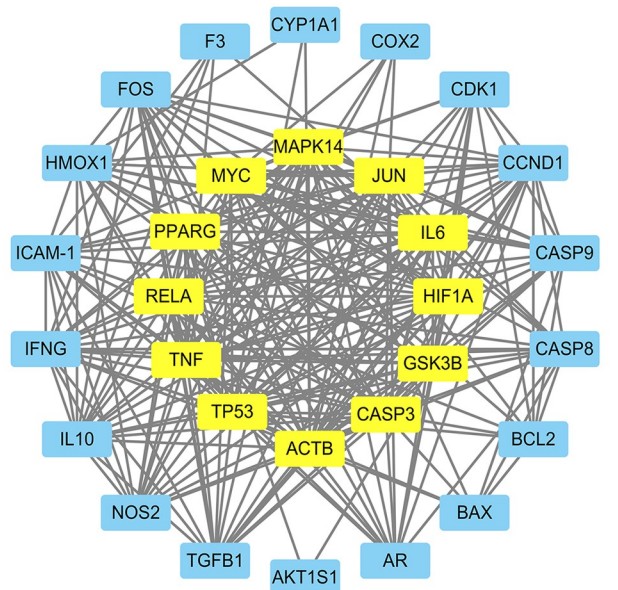

**Fig 2. Protein-protein interaction (PPI) analysis of FZJDS against PRRSV.** (A) PPI networks of 30 intersecting shared targets between FZJDS and PRRSV analyzed by STRING. (B) Protein interaction network of identified core targets related to the action of FZJDS against PRRSV by topology selection. Light blue nodes were regular targets while yellow nodes were core targets.

**Table 2. Hub proteins identified by cytoNCA.**

| Target | Degree | Betweenness | Closeness |
|---|---|---|---|
| ACTB | *25.0* | 49.47529882 | 0.878787879 |
| TP53 | 25.0 | 43.00693959 | 0.878787879 |
| CASP3 | 25.0 | 25.30145619 | 0.878787879 |
| TNF | 24.0 | 32.45018319 | 0.852941176 |
| MAPK14 | 24.0 | 23.87471626 | 0.852941176 |
| JUN | 24.0 | 20.63703153 | 0.852941176 |
| IL6 | 22.0 | 17.76169796 | 0.805555556 |
| HIFIA | 21.0 | 7.505629601 | 0.783783784 |
| MYC | 21.0 | 18.41436314 | 0.783783784 |
| RELA | 21.0 | 10.97528602 | 0.783783784 |
| GSK3B | 21.0 | 36.91209942 | 0.783783784 |
| PPARG | 20.0 | 28.90931291 | 0.763157895 |

and signal transduction pathways, as well as associated with important pathological processes such as apoptosis. The top 20 entries were selected according to the *P* value and are shown as a bubble diagram (**Fig 4B**). As shown in **Fig 4B**, the main signaling pathways included PI3K/Akt, JAK-STAT, NF-B, sphingolipid, Th1 and Th2 cell differentiation, and relaxin. The top 20 pathways with lower *P* and greater gene enrichment are listed in **Table 4**. Lastly, the herbs and 30 core targets were mapped onto the top 20 corresponding pathways, (**Fig 5**).

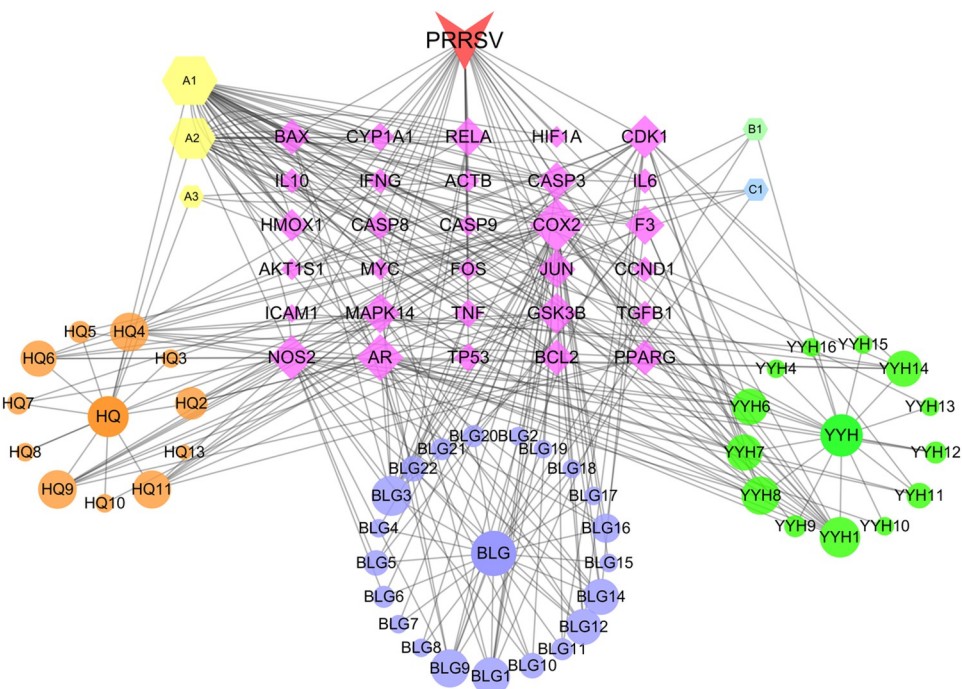

**Fig 3. Construction of network of 30 core targets of FZJDS (components) and viral disease (PRRSV).** The nodes denoting the compounds are shown as hexagons representing the intersecting ingredients of the three herbs. The names of the herbs are indicated by ellipses. Different colors represent different compounds; targets are indicated by purple diamonds, and red V's represent disease. Node size is shown in ascending order according to degree.

**Table 3. Top 15 active compounds and corresponding therapeutic targets of FZJDS.**

| Label | MOL ID | Name of compound | Degree | Names of targets |
|-------|--------|------------------|--------|------------------|
| A1 | MOL000098 | quercetin | 46 | COX-2,F3,CYP1A1,TGFB1,IFNG,JUN,CCND1,HIF1A,MYC,CASP9,CASP3,CASP8,IL10,HMOX1,FOS,RELA,TP53,BCL2,BAX,AR,IL6,GSK3B,TNF,PPARG |
| A2 | MOL000422 | kaempferol | 27 | GSK3B,AR,BAX,BCL2,CASP3,RELA,HMOX1,TNF,JUN,CYP1A1,PPARG,F3,COX-2,NOS2 |
| YYH1 | MOL000006 | luteolin | 13 | COX-2,IFNG,JUN,CCND1,CASP9,CASP3,IL10,HMOX1,RELA,TP53,AR,IL6 |
| BLG3 | MOL001689 | acacetin | 12 | COX-2,NOS2,AR,BAX,BCL2,CASP3,CASP8,RELA,TP53,TNF,CDK1 |
| HQ9 | MOL000392 | formononetin | 9 | COX-2,NOS2,GSK3B,AR,JUN,CDK1,MAPK14,PPARG |
| HQ4 | MOL000354 | isorhamnetin | 9 | COX-2,NOS2,GSK3B,AR,RELA,CDK1,MAPK14,PPARG |
| YYH8 | MOL004373 | Anhydroicaritin | 8 | COX-2,MAPK14,F3,CDK1,GSK3B,AR,NOS2 |
| BLG9 | MOL001767 | hydroxyindirubin | 8 | COX-2,NOS2,AR,GSK3B,PPARG,CDK1,MAPK14 |
| BLG1 | MOL000359 | beta-sitosterol | 8 | COX-2,BAX,BCL2,CASP3,CASP8,JUN,TGFB1 |
| HQ11 | MOL000417 | calycosin | 8 | COX-2,NOS2,GSK3B,AR,CDK1,MAPK14,PPARG |
| YYH14 | MOL004391 | 8-(3-methylbut-2-enyl)-2-phenyl-chromone | 7 | COX-2,MAPK14,CDK1,GSK3B,AR,NOS2 |
| YYH7 | MOL003542 | 8-isopentenyl-kaempferol | 7 | COX-2,MAPK14,F3,CDK1,GSK3B,AR |
| YYH6 | MOL003044 | Chryseriol | 7 | COX-2,MAPK14,CDK1,GSK3B,AR,NOS2 |
| BLG14 | MOL001793 | (E)-2-[(3-indole)cyanomethylene-]-3-indolinone | 7 | COX-2,NOS2,AR,GSK3BCDK1,MAPK14 |
| BLG12 | MOL001782 | (2Z)-2-(2-oxoindolin-3-ylidene)indolin-3-one | 7 | COX-2,NOS2,AR,GSK3B,CDK1,MAPK14 |

### 3.4. FZJDS cytotoxicity test on MARC-145 cells

In this study, it was noted that incubating MARC-145 cells with FZJDS under serum deprivation affected cell morphology. Furthermore, the cytotoxicity of FZJDS on MARC-145 cells was determined by CCK-8 assay. FZJDS did not significantly reduce MARC-145 viability at concentrations up to 16.00 mg/mL, but some cytotoxicity did occur above that (**Fig 6, Table 5**). These results demonstrate that FZJDS has minimal cytotoxicity on MARC-145 cells within the tested dose range accompany with a dose-dependent manner. FZJDS inhibited PRRSV in a dose-dependent manner and had a selectivity index (SI) greater than 10 (**Fig 6**).

### 3.5. FZJDS treatment suppressed PRRSV proliferation in MARC-145 cells

To determine the anti-PRRSV activity of FZJDS, MARC-145 cells were infected with PRRSV (100 $TCID_{50}$) for 1 h and then treated with different concentrations of FZJDS for 24, 36 and 48 h. As shown in (**Fig 7A and 7B**), treatment with FZJDS results in a significant dose-dependent and time-dependence reduction in PRRSV N mRNA levels both in MARC-145 cells. N mRNA expression, representing the PRRSV replication rate in the treated groups, was compared to the control group (**Fig 7A and 7B**). The N mRNA expression significantly decreased from 36 h.p.i. to 48 h.p.i. in MARC-145 cells, a 80% decrease the amount of PRRSV mRNA in the presence of 15.63 mg/mL FZJDS (**Fig 7A**). The expression level of N protein, evaluated by western blotting, decreases in proportion to the amount of FZJDS used in the treatment (**Fig 7C–7F**). Virus proliferation was analyzed from 24h to 48 h in the presence with 15.63 mg/mL FZJDS, the N protein expression levels only decrease in the 36 h and 48 h group (**Fig 7f**). Collectively, these results suggest that FZJDS treatment suppresses PRRSV infection.

### 3.6. FZJDS inhibits PI3K/Akt signaling pathway

In the present study, results of network pharmacology were verified by western blotting and qRT-PCR. MARC-145 cells were infected with PRRSV (100 TCID50) for 1 h and then treated

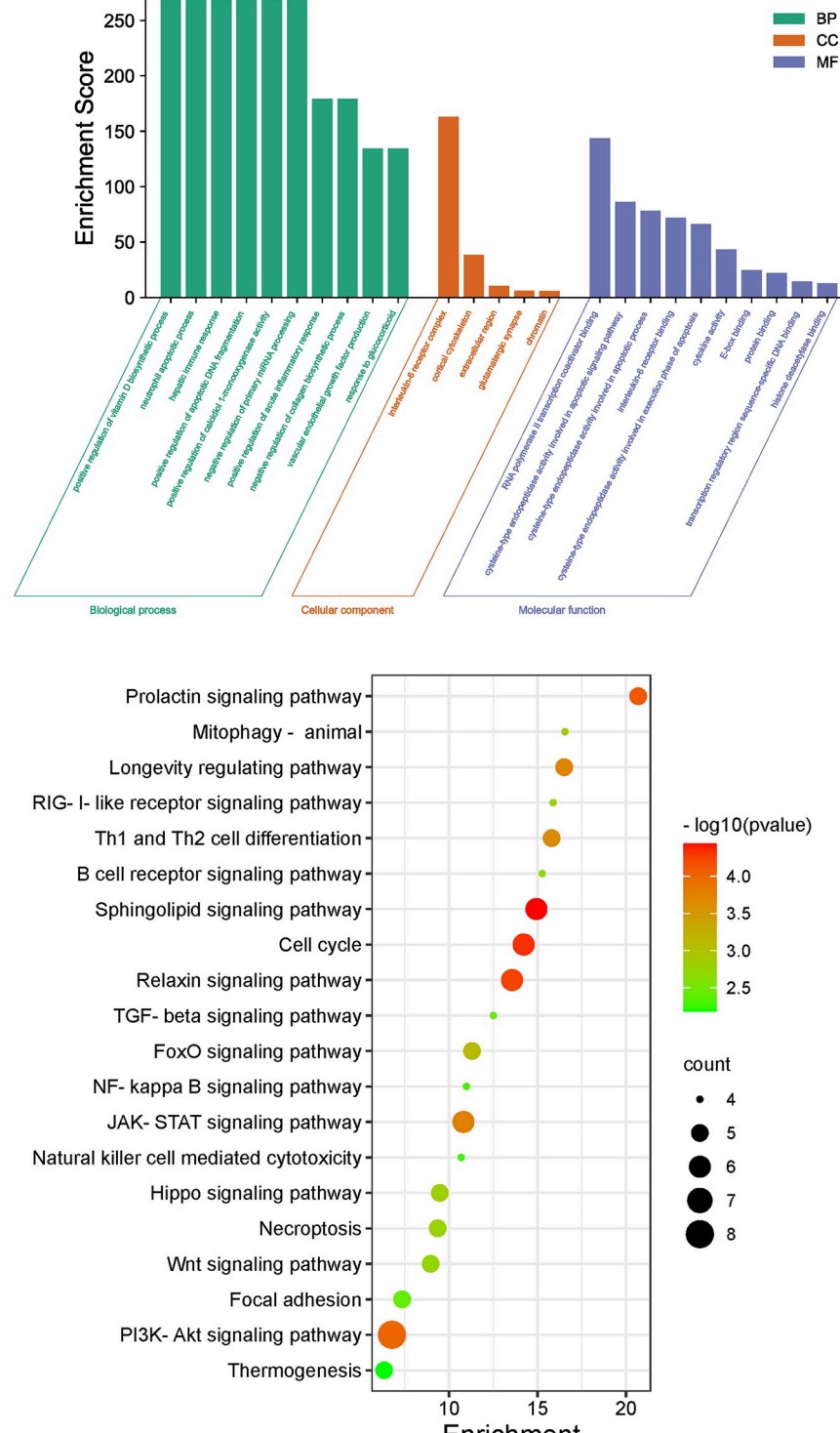

**Fig 4. Functional characterization of FZJDS against PRRSV intersecting genes.** (a) Gene ontology analysis of intersecting genes of FZJDS and PRRSV. (b) Kyoto Encyclopedia of Genes and Genomes (KEGG) pathways of intersecting genes between FZJDS and PRRSV.

**Table 4. Classification and information on top 20 KEGG pathways related to core targets.**

| Classification | Pathway | *P* | Gene count |
|---|---|---|---|
| Cellular processes | Cell cycle | 4.61E-05 | 6 |
| | Mitophagy—animal | 1.58E-03 | 4 |
| | Necroptosis | 1.63E-03 | 5 |
| | Focal adhesion | 3.92E-03 | 5 |
| Environmental information processing | Sphingolipid signaling pathway | 3.63E-05 | 6 |
| | PI3K-Akt signaling pathway | 1.02E-04 | 8 |
| | JAK-STAT signaling pathway | 1.69E-04 | 6 |
| | FoxO signaling pathway | 8.07E-04 | 5 |
| | Hippo signaling pathway | 1.55E-03 | 5 |
| | Wnt signaling pathway | 1.91E-03 | 5 |
| | TGF-β signaling pathway | 3.51E-03 | 4 |
| | NF-κB signaling pathway | 5.06E-03 | 4 |
| Organismal systems | Relaxin signaling pathway | 5.78E-05 | 6 |
| | Prolactin signaling pathway | 7.89E-05 | 5 |
| | Longevity regulating pathway | 1.90E-04 | 5 |
| | Th1 and Th2 cell differentiation | 2.26E-04 | 5 |
| | RIG-I-like receptor signaling pathway | 1.78E-03 | 4 |
| | B cell receptor signaling pathway | 1.99E-03 | 4 |
| | Natural killer cell-mediated cytotoxicity | 5.46E-03 | 4 |
| | Thermogenesis | 6.63E-03 | 5 |

with different concentrations of FZJDS for 48 h. As shown in **Fig 8**, FZJDS inhibited the increased phosphorylation of P65, JNK and IBa in MARC-145 cells compared to the negative control (**Fig 8C, 8F and 8I**). There was no change in expression of non-phosphorylated proteins. Results of qRT-PCR and western blotting indicated that expression of TLR4 and Bax were significantly decreased by FZJDS treatment (**Fig 8A and 8M**), while Bcl-2 expression was increased (**Fig 8L**). These findings support the proposition that FZJDS can exert an antiviral effect via the PI3K/Akt signaling pathway.

### 3.7. FZJDS inhibits PRRSV-induced apoptosis of MARC-145 cells

Our results showed that FZJDS inhibited expression of the apoptotic protein, Bax, and promoted the expression of the anti-apoptotic protein, Bcl-2, in MARC-145 cells. To investigate whether FZJDS inhibited PRRSV-induced apoptosis of MARC-145 cells, PRRSV-infected cells were treated with different concentrations of FZJDS and the apoptosis rate was measured by flow cytometry 48 h later. The results showed that apoptosis in MARC-145 cells infected with PRRSV was significantly lower with FZJDS treatment than in the untreated group. The apoptosis rate in infected MARC-145 cells decreased with increasing FZJDS concentration ($P < 0.01$; **Fig 9**).

To explore whether FZJDS exerted anti-apoptotic effects by affecting JNK, we compared the levels of JNK protein after inhibitor treatment. MARC-145 cells were first infected with PRRSV and then treated with an inhibitor to block the phosphorylation of JNK. After 48 h, flow cytometry was used to measure the effects of JNK signaling pathway inhibition on apoptosis induced by PRRSV. The results showed that compared with the control group, the apoptosis rate was markedly reduced at an FZJDS concentration of 7.81 mg/mL, with a maximum at a dose of 15.63 mg/ mL ($P<0.0001$; **Fig 10A–10E**). Western blotting results showed that the JNK inhibitor (10 μM) significantly reduced the levels of P-JNK ($P < 0.01$; **Fig 10G**). When

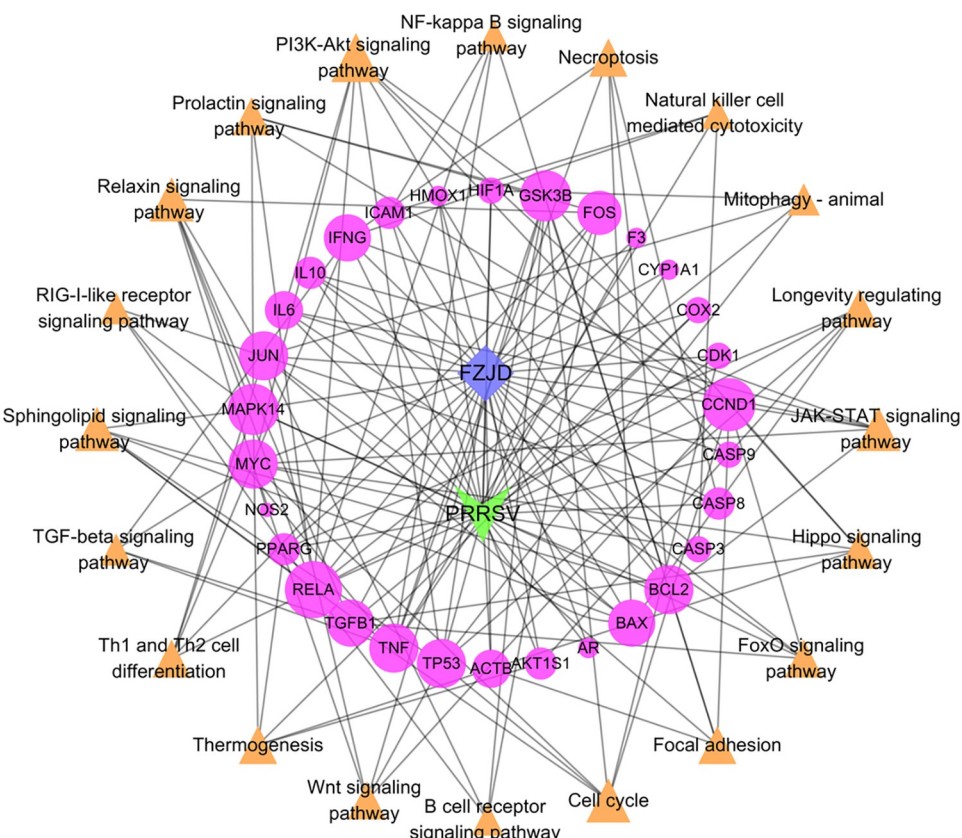

**Fig 5. Network illustrating the interaction among herbs, disease, core targets and corresponding pathways.** The blue diamonds indicate FZJDS, while the green V's represent PRRSV. The purple ellipses indicate the pathway-related targets, while, the orange triangles show the corresponding pathway. Node size is in ascending order according to degree.

JNK activation was inhibited, the apoptosis rate induced by PRRSV was significantly decreased ($P < 0.01$; **Fig 10F and 10G**) in a dose-dependent manner. The experimental results supported the hypothesis that FZJDS inhibited apoptosis by regulating JNK activation.

## 4. Discussion

Although PRRSV has been well controlled in China, side effects and antimicrobial resistance has been caused by the inappropriate application of drugs. Traditional Chinese medicine is a complementary and alternative approach involving many components acting via multiple mechanisms and targets. In addition to the antiviral function, the medications also regulate the host immune response to provide a synergistic effect. The TCM philosophy encourages holistic treatment with a systematic approach that leads to better therapeutic outcomes and fewer side effects. To study this process in more detail, we employed a systems pharmacology method to elucidate the potential pharmacological mechanism of FJZD's effect on PRRSV infection.

In this study, we conducted a systematic study using a combination of network pharmacology and experimental verification. The several compounds in FZJDS can affect multiple targets. Overlapping targets were identified for the different compounds, and it was apparent that FZJDS exerts an anti-PRRSV action through the synergistic effects of its compounds. Our results showed that quercetin, kaempferol, luteolin, acacetin and formononetin were the top

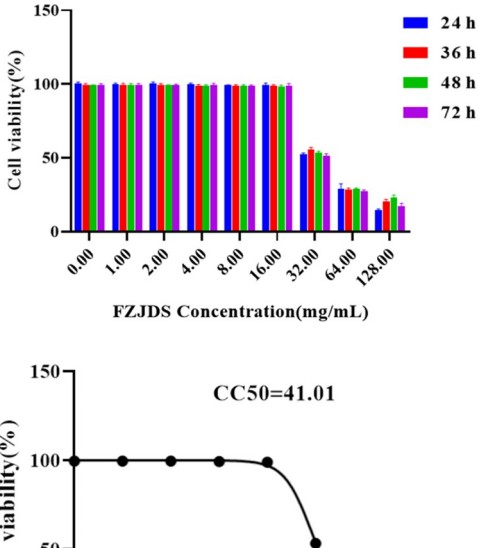

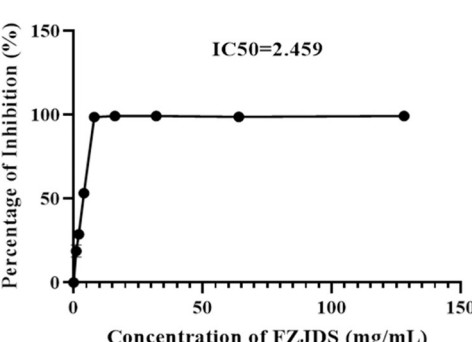

**Fig 6. Cytotoxicity of FZJDS on MARC-145 cells.** (a) Cell viability of FZJDS. (b) CC50 curves of FZJDS. (c) IC50 curves of FZJDS.

five active ingredients affecting the intersecting targets. A previous study showed that quercetin significantly suppressed porcine epidemic diarrhea virus (PEDV) infection at non-cytotoxic concentrations, by exerting an inhibitory effect on PEDV 3CLpro [11]. Another study reported that BanLanGen exhibited significant inhibitory activity against various subtypes of avian influenza viruses (IC50 = 0.39–4.3 mg/ml) and also inhibited degradation of IκBα and production of PGE2, NO, and IL-6 in LPS-stimulated RAW264.7 monocytes [17]. Western blotting was performed to measure the expression of COX-2, cleaved caspase-3, and peroxisome proliferator-activated receptor-γ (PPAR-γ), indicated that formononetin could protect against ox-LDL-induced inflammatory reactions, oxidative stress, and apoptosis in human vascular endothelial cells, HUVECs [18]. YinYangHuo enhanced the immune system by improving the response of spleen antibody-forming cells to near normal, upregulating lymphocyte proliferation and promoting the recovery of *IL-2* production [19]. In viral and post-viral entry experiments, luteolin significantly blocked foot-and-mouth-disease virus (FMDV) growth at

**Table 5. Cytotoxicity of FZJDS on MARC-145 cells (mean ± SD, n = 6).**

| FZJDS | Absorbance at 450 nm | | | |
|---|---|---|---|---|
| | 24h | 36h | 48h | 72h |
| Control | 0.85±0.035 | 1.00±0.045[A] | 1.04±0.036[B] | 0.97±0.044 |
| 128.00 mg/mL | 0.33±0.042[d] | 0.33±0.035[d] | 0.37±0.052[d] | 0.22±0.054[d] |
| 64.00 mg/mL | 0.35±0.034[d] | 0.29±0.037[d] | 0.38±0.032[d] | 0.38±0.048[d] |
| 32.00 mg/mL | 0.67±0.046[b] | 0.800±0.054[b] | 0.90±0.041[aB] | 0.80±0.055[a] |
| 16.00 mg/mL | 0.90±0.062 | 1.05±0.043 | 1.08±0.058[A] | 1.02±0.040 |
| 8.00 mg/mL | 0.82±0.047 | 1.02±0.045[A] | 1.10±0.039[B] | 1.03±0.053[B] |
| 4.00 mg/mL | 0.85±0.049 | 1.06±0.047[B] | 1.04±0.039[B] | 1.05±0.029[B] |
| 2.00 mg/mL | 0.85±0.054 | 0.97±0.049 | 1.11±0.065[B] | 1.05±0.031[A] |
| 1.00 mg/mL | 0.80±0.046 | 0.88±0.046 | 1.02±0.035[B] | 1.04±0.051[B] |

Note: Different lowercase letters in the same column (culture time) denote significant differences among concentrations. Different uppercase letters in the same row denote significant differences among culture times at the same medication concentration.

low 50% effective concentrations (EC50), and could reduce the viral load as determined by RT-qPCR [20]. Taken together, these results support the idea that the active compounds in FZJDS have synergistic antiviral effects.

In the present study, GO enrichment and KEGG pathways enrichment analysis were carried out to explore the multi-dimensional pharmacological mechanism of FZJDS. After

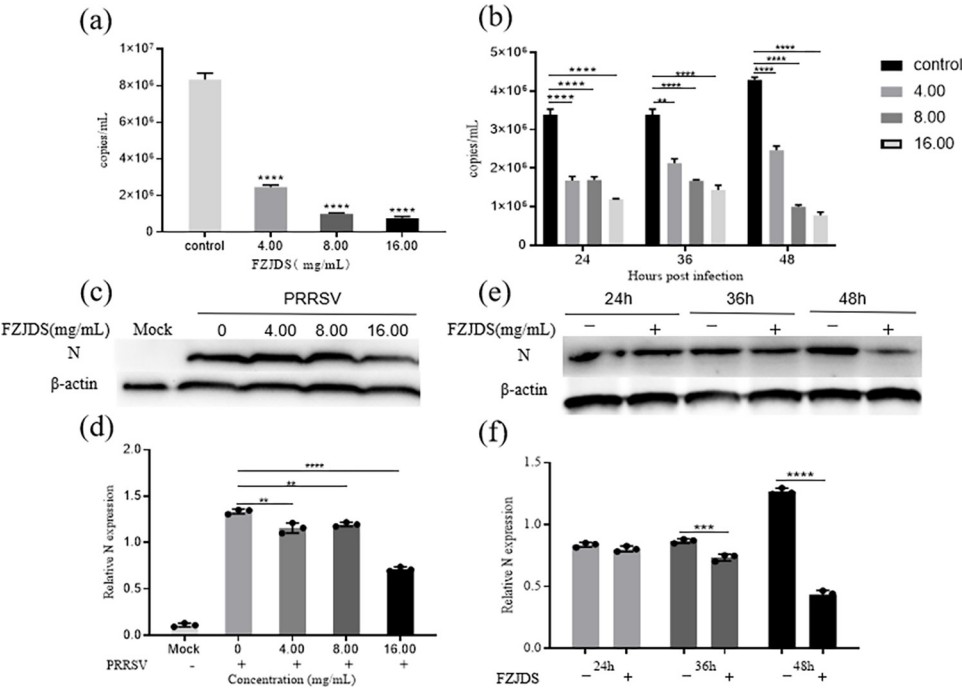

**Fig 7. Effect of FZJDS on the growth of porcine reproductive and respiratory syndrome virus (PRRSV) in MARC-145 cells.** (a) Effect of FZJDS dose on mRNA expression of the PRRSV N gene as measured by qRT-PCR. (b) Effect of time of FZJDS exposure on mRNA expression of PRRSV N gene as measured by qRT-PCR. (c) Effect of FZJDS dose on PRRSV N protein expression as determined by western blotting. (d) Results of greyscale analysis of Fig 7C. (e) Effect of time of FZJDS exposure on PRRSV N protein expression as measured by western blotting. (f) Results of greyscale analysis of Fig 7E. Data shown in a-f are means ± SEM from four individual wells per group in one experiment. * $P<0.05$, ** $P<0.01$, *** $P<0.001$, **** $P<0.0001$ vs. control group.

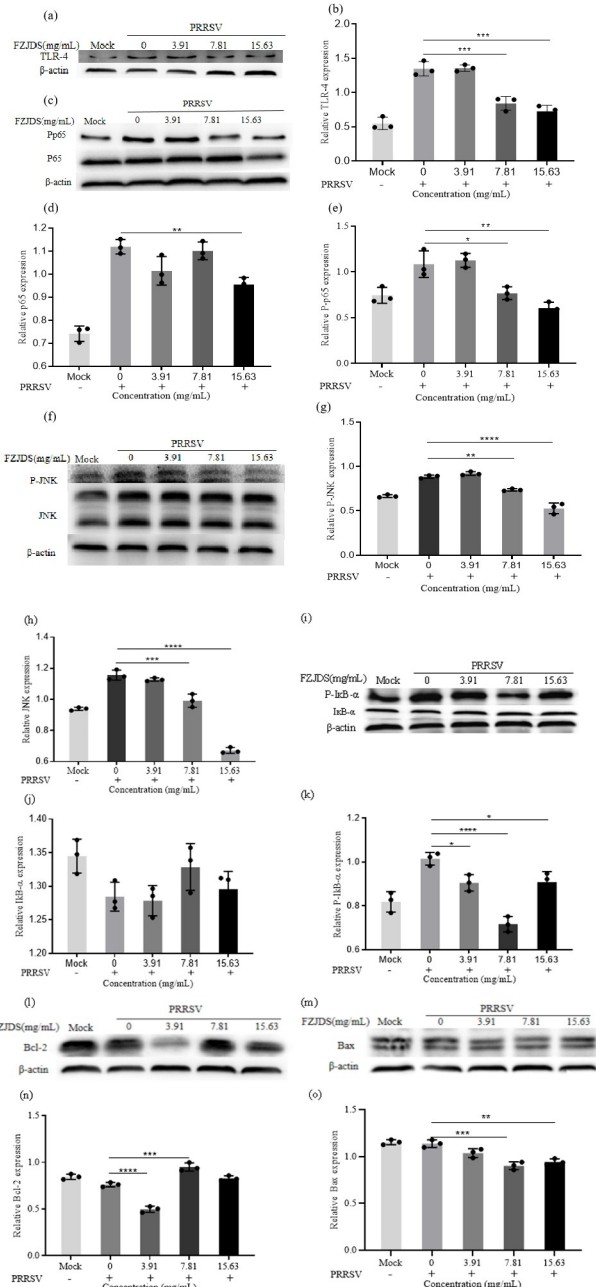

**Fig 8. Effect of FZJDS on PI3K/Akt pathway in MARC-145 cells.** (a) Western blotting analysis shows the level of TLR-4 in MARC-145 cells infected with PRRSV (100 TCID50) after treatment with FZJDS. (b) Quantification of protein levels from the western blotting in panel a. (c) Western blotting analysis shows the level of P-p65 and P65 in MARC-145 cells infected with PRRSV (100 TCID50) after treatment with FZJDS. (d, e) Quantification of protein levels from the western blotting in panel c. (f) Western blotting analysis shows the level of P-JNK and JNK in MARC-145 cells infected with PRRSV (100 TCID50) after treatment with FZJDS. (g, h) Quantification of protein levels from the western blotting in panel c. (f) Western blotting analysis shows the level of P-JNK and JNK in MARC-145 cells infected with PRRSV (100 TCID50) after treatment with FZJDS. (g,h) Quantification of protein levels from the western blotting in panel f. (i) Western blotting analysis shows the level of P-IkBa and IkBa in MARC-145 cells infected with PRRSV (100 TCID50) after treatment with FZJDS. (j,k) Quantification of protein levels from the western blotting in panel i. (l) Western blotting analysis shows the level of Bcl-2 in MARC-145 cells infected with PRRSV (100 TCID50) after treatment with FZJDS. (n) Quantification of protein levels from the western blotting in panel l. (m) Western blotting analysis shows the levels of Bax in MARC-145 cells infected with PRRSV (100 TCID50) after treatment with FZJDS. (o) Quantification of protein levels from the western blotting in panel m. For b, d, e, g, h, j, k, n and o, results are means ± SEM from three independent experiments, each of which was performed in triplicate. $*P < 0.05$; $**P < 0.01$, $***P < 0.001$ and $****P < 0.0001$.

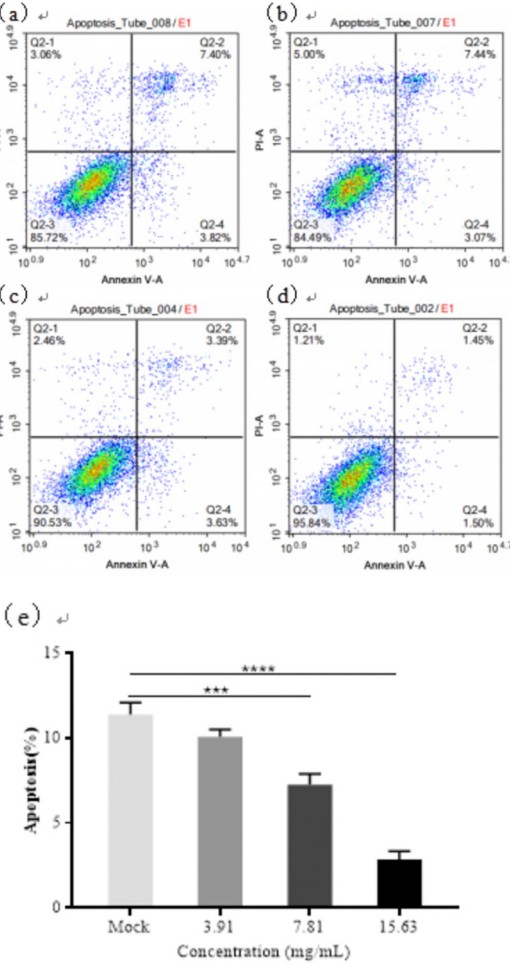

**Fig 9. Effect of FZJDS on apoptosis rate in PRRSV-infected MARC-145 cells.** The apoptosis rate in PRRSV-infected MARC-145 cells treated with different FZJDS concentrations was determined by flow cytometry with annexin V/ propidium iodide (PI) dual staining (a,b,c,d). The percentage of apoptotic cells is shown in (e). $^*P < 0.05$; $^{**}P < 0.01$, $^{***}P < 0.001$, $^{****}P < 0.0001$.

removing human diseases, we analyzed the pathways with more annotation genes and a lower *P*-value. Among these pathways, the PI3K-Akt signaling pathway, sphingolipid signaling pathway and JAK-STAT signaling pathway are the main pathways with the highest enrichment scores. Among these pathways, FZJDS mainly interfered with the occurrence and development of PRRSV through the PI3K-Akt signaling pathway. This was in agreement with other research suggesting that angiotensin-converting enzyme 2 (ACE2)-mediated SARS-CoV-2 spike pseudovirions (SCV-2-S) infection induced autophagy and apoptosis in human bronchial epithelial and microvascular endothelial cells. SCV-2-S inhibited the PI3K/AKT/mTOR pathway by upregulating intracellular reactive oxygen species (ROS) levels, thus promoting the autophagic response [21]. One study demonstrated that involvement of the PI3K/AKT/mTOR pathway in megakaryocyte development and maturation was confirmed with the use of specific inhibitors in Dengue virus-infected MEG-01 cells [22]. The expression of NF-κB p65, p-NF-κB and p65 was increased with favipiravir treatment, implying that the antiviral effectivity of favipiravir against peste des petits ruminants virus (PPRV) is mediated by the JAK/STAT and PI3K/AKT pathways [23]. A recent genome-wide mRNA and long non-coding RNA (lncRNA) search

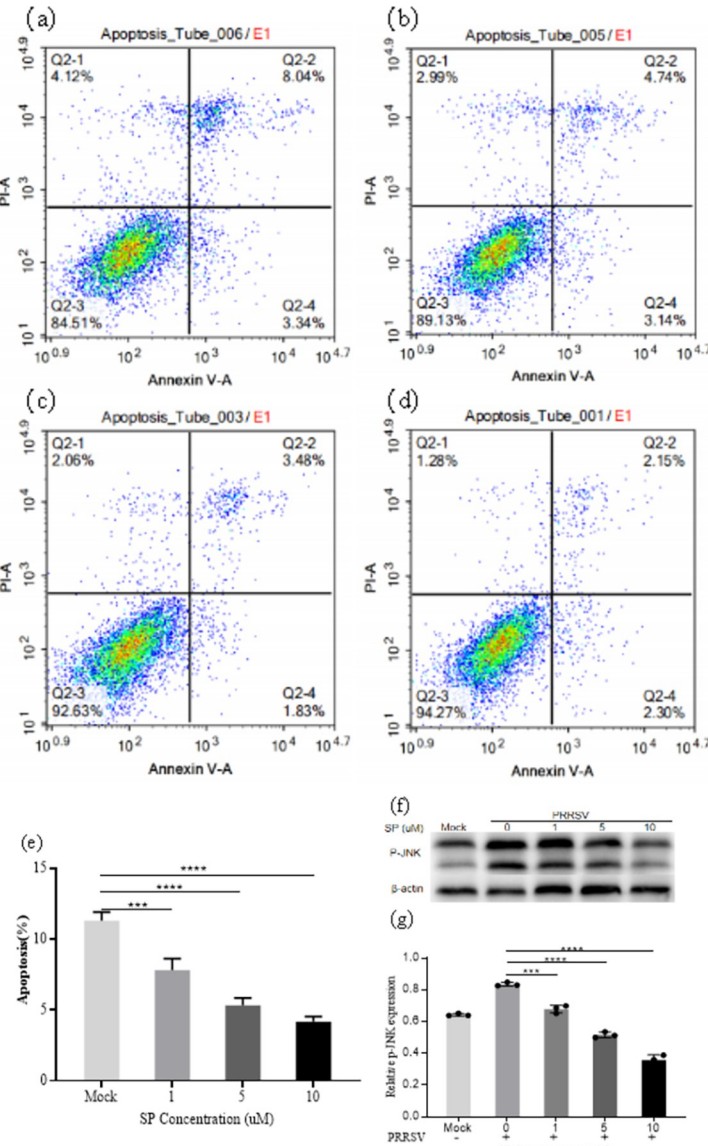

**Fig 10. FZJDS inhibits activation of JNK in MARC-145 cells.** The effect of P-JNK inhibition on apoptosis in MARC-145 cells induced by PRRSV was determined by flow cytometry (a,b,c,d). The percentage of apoptotic cells was also determined (e). (f) Western blotting analysis shows the levels of P-JNK after inhibitor treatment in MARC-145 cells. (g) Quantification of P-JNK protein levels from the western blotting data in panel f. For e and g, results are shown as the mean ± SEM from three independent experiments, each of which was performed in triplicate. *$P < 0.05$; **$P < 0.01$, ***$P < 0.001$, ****$P < 0.0001$.

and analysis in PRRSV-infected PTR2 showed that PRRSV infection could inhibit the PI3K/AKT/mTOR pathway and trigger apoptosis [24]. Another study determined the phosphorylation status of Akt during infection of MARC-145 cells with the highly pathogenic PRRSV strain, HuN4, and provided new evidence of a novel role for the PI3K/Akt pathway in PRRSV infection [25]. These evidence showed that the PI3K-AKT signaling pathway played an important role in regulating apoptosis FZJDS has enormous potential for application in animal husbandry due to its remarkable immunomodulatory effects and antiviral activity.

We found that Bax, Bcl-2, TLR4 and p53 as the core target proteins of FZJDS in response to PRRSV infection of MARC-145 cells was also predicted from the network pharmacology analysis. As previously described by Lenka Kavanová, (2017), monocyte derived macrophages infected with PRRSV could significantly increase the mRNA expression of pro-apoptotic genes, such as *bad*, *Bax* and *p53* [26]. Recent evidence suggests that PRRSV infection aggravated the morphological depletion of tight junction proteins and increased *IL-1β*, *IL-6*, *IL-8* and *TNF-α* expression by activating the NF-κB signaling pathway in the jejunum [27]. Another previous study showed specific host transcriptome differences in porcine alveolar macrophages (PAMs) at seven days post-challenge. In Tongcheng (TC) pigs, 549 specific differentially expressed genes (DEGs) were identified, including VAV2, Bcl-2 and Bax, which were enriched in activation of leukocyte extravasation and suppression of apoptosis [28]. Mice infected with adapted influenza A virus (IAV) H1N1 strain A/Font Monmouth were treated with protocatechuic acid (PCA). PCA reduced infiltration of immune cells and cytokine levels in the lung, as well as suppressed H1N1-induced TLR4/NF-κB activation [29]. These results are in agreement with our network pharmacology results showing the core predicted targets of FZJDS in treating PRRSV-induced apoptosis. We also observed the effects on apoptosis related genes, Bax, Bcl-2, TLR-4, JNK and IBa, by real-time qRT-PCR and western blotting. In addition, a recent research report showed that PRRSV replicated in the lungs and small intestine and that the PRRSV N protein was detected in the lung interstitial and jejunal mucosa [27]. We measured the expression level of N protein by western blotting in our present study, and showed that it decreased in proportion to the FZJDS dose used in the treatment.

We also found that the *JNK* inhibitor, SP600125, inhibited PRRSV-mediated apoptosis by activation of the downstream protein, IBa. The inhibition of JNK not only significantly inhibited virus-induced autophagosome formation, but also suppressed replication of the corresponding virus [30]. In this study, FZJDS decreased phosphorylation of P65, JNK, and IBa in MARC-145 cells, and decreased the expression of TLR4 and Bax. In previous studies immunoprecipitation assays have revealed that BEFV disrupted Beclin 1 and Bcl-2 interaction by JNK-mediated Bcl-2 phosphorylation, thereby activating autophagy [30], which proves that JNK and Bcl-2 are core proteins affecting PRRSV. Activated JNK translocates to the mitochondria where it phosphorylates Bcl-2/Bcl-xL and antagonizes the anti-apoptotic activity of Bcl-2/Bcl-xL. Moreover, JNK expression leads to the release of cytochrome C via a bid-bax-dependent mechanism, activating caspase 9 and caspase 3 and inducing apoptosis [31], consistent with our findings. In view of the above results, we speculate that the active compounds of FJZDS target genes related to the pathways that inhibit apoptosis such as the PI3K-AKT signaling pathway, and FZJDS plays an important role in protecting MARC-145 cells from apoptosis through PI3K-AKT signaling pathway, suggesting that FZJDS exerted its antiviral effects through inhibiting multiple cytokines and proteins.

In summary, our findings suggest that the protective mechanism of FZJDS against PRRSV is related to the activation of the PI3K/AKT signaling pathway through TLR4, Bax, JNK, Bcl-2 and IBa, and that the main active ingredients are quercetin, kaempferol and luteolin. This is the first study that comprehensively analyzed the main compounds, targets, and pathways of FZJDS in treating PRRSV utilizing a network pharmacology approach together with experimental validation, which supports FZJDS as an alternative therapy for treating PRRSV infection.

## Supporting information

**S1 Data.**
(ZIP)

**S1 Graphical abstract.**
(TIF)

## Acknowledgments

The authors would like to thank Beijing Key Laboraory of Traditional Chinese Veterinary Medicine and Beijng Key Laboratory of Dairy Cow Nutrition, for assisting with this article and technical support.

## Author Contributions

**Data curation:** Kexin Chang.

**Funding acquisition:** Yonghong Zhang.

**Methodology:** Qiong Wu.

**Project administration:** Defeng Cui.

**Resources:** Yonghong Zhang, Defeng Cui.

**Software:** Kexin Chang.

**Supervision:** Hua Zhang, Qiong Wu, Jinjin Tong, Defeng Cui.

**Validation:** Kuangshi Fan, Le Wang, Hongcen Chen.

**Visualization:** Kexin Chang.

**Writing – original draft:** Kexin Chang, Kuangshi Fan.

**Writing – review & editing:** Hua Zhang, Jinjin Tong.

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
