## [Decision Letter · Decision Letter 0]

2 Sep 2022

PONE-D-22-18907

Fuzhengjiedu San inhibits porcine reproductive and respiratory syndrome virus by activating the PI3K/AKT pathway

PLOS ONE

Dear Dr. Chang,

Thank you for submitting your manuscript to PLOS ONE. After careful consideration, we feel that it has merit but does not fully meet PLOS ONE’s publication criteria as it currently stands. Therefore, we invite you to submit a revised version of the manuscript that addresses the points raised during the review process.

We look forward to receiving your revised manuscript.

Kind regards,

Ahmad Salimi

Academic Editor

PLOS ONE

Journal Requirements:

2. We noted that some of the raw blot images provided do not appear to match the images shown in the figures (e.g. Fig6c and N¦¦¦+.tif). Several raw images provided include more rows/lanes than are shown in the figure (e.g. Fig7c and p65.tif). Please comment on this and ensure all the raw images match their corresponding figures in the manuscript. Please also provide the lane loading information for all raw blot data files, either by adding labels directly in the image files or by providing a supporting file that lists the lane loading information for all blot files.

   "This study was financially supported by the Swine Innovation Team Project of the Beijing Modern Agricultural Industrial Technology System (BAIC02) ."

 "This study was financially supported by the Swine Innovation Team Project of the Beijing Modern Agricultural Industrial Technology System (BAIC02).The funders had no role in study design, data collection and analysis, decision to publish, or preparation of the manuscript."

Reviewers' comments:

Reviewer's Responses to Questions

**Comments to the Author**

1. Is the manuscript technically sound, and do the data support the conclusions?

Reviewer #1: Yes

Reviewer #2: Partly

2. Has the statistical analysis been performed appropriately and rigorously? 

Reviewer #1: Yes

Reviewer #2: Yes

3. Have the authors made all data underlying the findings in their manuscript fully available?

Reviewer #1: Yes

Reviewer #2: No

4. Is the manuscript presented in an intelligible fashion and written in standard English?

Reviewer #1: Yes

Reviewer #2: No

5. Review Comments to the Author

Reviewer #1: In order to investigate the antiviral effect of Porcine reproductive and respiratory syndrome virus (PRRSV) , this manuscript uses a network pharmacological approach and finds that some key biomolecules are associated with the antiviral effect of PRRSV.

on the basis of network pharmacology research, using cytological method, respectively, to drug toxicity, antiviral activity, the study found that some of the key effect on molecular level, The anti - apoptosis effect has been studied and many results have been obtained. These results suggest that the anti-PRRSV effect of FZJDS is mainly through the PI3K/AKT signaling pathway, which provides some information on the molecular basis of the antiviral effect of FZJDS.

In this study, the method of network pharmacology was comprehensively introduced, which made the study more reproducible, which was the obvious advantage of this study.

There are still some problems with this manuscript that deserve serious consideration by the authors.

Although network pharmacology provides important information about molecular interaction, it is difficult to make a comparative analysis between network pharmacology research and cytological research because of the huge difference between kidney cells （MARC‐ 145 cells derived from African green monkey kidney）selected and immune cells mainly from antiviral infection in cytological experiments.

The initial concentration of the drug was 125mg, because the use of multiple dilution resulted in the subsequent dose showing two decimal places（3.91，7.81mg/L，I do not think 3.91 is better than 4.0 here.）. This accuracy was not helpful for the analysis and understanding of its pharmacological effect, and the authors should reduce the number of decimal places. It is recommended to choose an initial dose of 128 mg, so that all dilutions are rounded and the numbers are tidy. Research is a very personal job and requires neat habits. (3.4. FZJDS cytotoxicity test on MarC-145 cells)

In order to determine the antiviral effect of the drug, an antiviral drug should also be used as a positive control, which is relatively easy to achieve for cell studies and can be more clearly illustrated. (3.5, Figure 6)

The effects of FZJDS on uninfected cells were studied, but these effects were not analyzed. These effects are also necessary to analyze drug effects, so as to avoid non-specific concomitant effects. (3.6)

There is no need to discuss the results in the results section. These should be in the discussion section of the paper. Line 413 The Experimental Results supported The hypothesis that FZJDS Inhibited Apoptosis by regulating JNK Activation.)

To reinforce the evidence for the conclusion of this manuscript, the antiviral effect of pJNK inhibitors could be analyzed.

Reviewer #2: The manuscript by Chang et al reported that Fuzhengjiedu San (FZJDS), has been extensively used to treat viral diseases in pigs. Quercetin, kaempferol, and luteolin were the main active compounds of FZJDS. FZJDS significantly reduced the expression of P65, JNK, TLR4, N protein, Bax and IĸBa in MARC-145 cells, and increased the expression of Bcl-2, consistent with network pharmacology results. FZJDS has significant antiviral activity through its effects on the PI3K/AKT signaling pathway. We conclude that FZJDS is a promising candidate herbal formulation for treating PRRSV and deserves further investigation. Nevertheless, there are many Major Compulsory Revisions to be faced before the manuscript could be considered for publication.

The content of this paper is complex and the logic is not clear, thus the manuscript is difficult to follow.

1. The description of Fuzhengjiedu San is not appropriate

2. The antiviral effects of FZJDS were only tested in Marc-145 cell, which is originated from green monkey. The natural target cell of PRRSV is PAM, the authors should use PAM to perform assays to demonstrate the anti-PRRSV activity of FZJDS.

3. The cytotoxicity of FZJDS on Marc-145 should be shown in graph rather than table. Further, the authors should calculate the CC50 of the drug.

4. In Figure 6C and E, the inhibitory effect of FZJDS on PRRSV replication is very weak, indicating that the drug may not be used for PRRSV control.

5. The authors also should calculate the IC50, which is indispensable for antiviral drug research.

6. The level of p-p65 in mock treatment is very high and is almost comparable to that of PRRSV infection groups. Commonly the p-p65 level in mock group is very low or cannot be detectable. This phenomenon is also observed in the level of p-JNK and p-IkBalpha. The data are questionable.

7. The data of bioinformatics shown in Figures 1-5 are too complex and difficult to follow, which makes the logic of the paper not clear.

8. Quality of written English: Not suitable for publication unless extensively edited.

6. PLOS authors have the option to publish the peer review history of their article (what does this mean?). If published, this will include your full peer review and any attached files.

Reviewer #1: No

Reviewer #2: No

---

## [Author Response · Author response to Decision Letter 0]

26 Oct 2022

Dear Editors and Reviewers,

We highly appreciate the detailed and valuable comments of the reviewers on our manuscript entitled “Fuzhengjiedu San inhibits porcine reproductive and respiratory syndrome virus by activating the PI3K/AKT pathway” (ID: PONE-D-22-18907). As below, on behalf of my co-authors, I would like to clarify all the points raised by the reviewers. And we hope the reviewers and the editors will be satisfied with our responses to the comments and the revisions for the original manuscript.

The main corrections in the paper and the responds to the reviewer’s comments are as flowing:

Responds to the reviewer’s comments:

Reviewer #1:

1.Although network pharmacology provides important information about molecular interaction, it is difficult to make a comparative analysis between network pharmacology research and cytological research because of the huge difference between kidney cells （MARC‐ 145 cells derived from African green monkey kidney）selected and immune cells mainly from antiviral infection in cytological experiments.

Ans: Thank you for your comment. In the present study, we selected sus scrofa species during the network pharmacologic analysis, besides previous studies have been widely elucidated that the mechanism of PRRSV infection on swine by the MARC‐145 cell line. In a study reported from Xun Wang et al found that tea polyphenol effectively inhibited PRRSV infection in Marc-145 cells by suppressing the stages of viral attachment, internalization, replication and release (Xun Wang et al. 2021). In line with previous publications showed that whole-genome transcriptome analysis of PRRSV-infected Marc-145 cells, which provides insight into the dynamics of host gene regulation during PRRSV infection (Ying Wei et al. 2018), as well as proved that PRRSV induced micropinocytosis via TIM-1 in MARC-145 cells (Xin Wei et al. 2020). Moreover, a proteomic study revealed that PRRSV virions incorporate a variety of simian proteins after replication in MARC-145 cells, which raises the intriguing possibility that cross-neutralization of PRRSV strains may, in particular, be mediated by antibodies recognizing simian antigen (David Goldeck et al. 2019). Previous findings suggest that the role of the plasma membrane cholesterol in PRRSV infection of MARC-145 cells was investigated by Ying Sun et al (2011). While knockdown of DHX36 and its adaptor myeloid differentiation primary response gene 88 (MyD88) by small-interfering RNA in MARC-145 cells significantly reduced NF-kB activation and pro-inflammatory cytokine expression after PRRSV infection (Huiyuan Jing et al. 2017). Therefore, we used MARC‐145 cells in the present study for the cytological research validation.

We all totally agree with immune cells used to clarify the PRRSV infection mechanisms, rather than only either the MARC-145 cells or the immune effects. In the further research ,we'll both use immune cells for the mechanism study. Thanks for your suggestions again.

[1] Wang X, Dong W, Zhang X, Zhu Z, Chen Y, Liu X, Guo C. Antiviral Mechanism of Tea Polyphenols against Porcine Reproductive and Respiratory Syndrome Virus. Pathogens. 2021 Feb 13;10(2):202. doi: 10.3390/pathogens10020202. PMID: 33668502; PMCID: PMC7917843.

[2] Wei Y, Li J, Zhang Y, Xue C, Cao Y. Tandem 3' UTR Patterns and Gene Expression Profiles of Marc-145 Cells During PRRSV Infection. Virol Sin. 2018 Aug;33(4):335-344. doi: 10.1007/s12250-018-0045-y. Epub 2018 Aug 1. PMID: 30069823; PMCID: PMC6178098.

[3] Wei X, Li R, Qiao S, Chen XX, Xing G, Zhang G. Porcine Reproductive and Respiratory Syndrome Virus Utilizes Viral Apoptotic Mimicry as an Alternative Pathway To Infect Host Cells. J Virol. 2020 Aug 17;94(17):e00709-20. doi: 10.1128/JVI.00709-20. PMID: 32522856; PMCID: PMC7431799.

[4] Goldeck D, Perry DM, Hayes JWP, Johnson LPM, Young JE, Roychoudhury P, McLuskey EL, Moffat K, Bakker AQ, Kwakkenbos MJ, Frossard JP, Rowland RRR, Murtaugh MP, Graham SP. Establishment of Systems to Enable Isolation of Porcine Monoclonal Antibodies Broadly Neutralizing the Porcine Reproductive and Respiratory Syndrome Virus. Front Immunol. 2019 Mar 27;10:572. doi: 10.3389/fimmu.2019.00572. PMID: 30972067; PMCID: PMC6445960.

[5] Sun Y, Xiao S, Wang D, Luo R, Li B, Chen H, Fang L. Cellular membrane cholesterol is required for porcine reproductive and respiratory syndrome virus entry and release in MARC-145 cells. Sci China Life Sci. 2011 Nov;54(11):1011-8. doi: 10.1007/s11427-011-4236-0. Epub 2011 Dec 16. PMID: 22173307; PMCID: PMC7088586.

[6] Jing H, Zhou Y, Fang L, Ding Z, Wang D, Ke W, Chen H, Xiao S. DExD/H-Box Helicase 36 Signaling via Myeloid Differentiation Primary Response Gene 88 Contributes to NF-κB Activation to Type 2 Porcine Reproductive and Respiratory Syndrome Virus Infection. Front Immunol. 2017 Oct 23;8:1365. doi: 10.3389/fimmu.2017.01365. PMID: 29123520; PMCID: PMC5662876.

2.The initial concentration of the drug was 125 mg, because the use of multiple dilution resulted in the subsequent dose showing two decimal places（3.91，7.81mg/L，I do not think 3.91 is better than 4.0 here.）. This accuracy was not helpful for the analysis and understanding of its pharmacological effect, and the authors should reduce the number of decimal places. It is recommended to choose an initial dose of 128 mg, so that all dilutions are rounded and the numbers are tidy. Research is a very personal job and requires neat habits.

Ans: Thank you for your suggestion. It has been revised.

3.In order to determine the antiviral effect of the drug, an antiviral drug should also be used as a positive control, which is relatively easy to achieve for cell studies and can be more clearly illustrated. (3.5, Figure 6)

Ans: Thank you for your suggestion. Previous study by Liu Lan et al. (2016), Ma Xia et al.(2015) has proved that FZJDS plays an important role in anti-infectious Bursal Disease Virus(IBDV). Moreover, Guangwen Dai et al. (2017) described that FZJDS has a certain cure rate in pigs with clinical infection of PRRSV, Porcine circovirus type 2 (PCV2) and Pseudorabies virus (PRV). Therefore, in the present study we only test the different concentration of FZJDS effect on PRRSV infection cell model, which would like to select the best and safest concentration by the cell viability. In particular, in previous studies it has been confirmed by Zhengtu Li et al. (2017) found that the polysaccharide extract from Radix isatidis root can be used as an adjunct to antiviral therapy for the IAV infection and LH-C might be an option for treating influenza virus infection(Yuewen Ding, 2017) without positive control. Therefore, no positive control is used in this study. We sincerely grateful all the reviewers' comment on the positive control group in the experiment design and will adding the positive controls in the following studies.

[1] Liu Lan, Li Shanshan, Wang Yafang, Peng Jinshan, Guo Zhihong, Zhang Junru, He Cheng. Clinical Efficacy of Fuzheng Jiedu Granules on Chickens Infected by IBDV [J].Chinese Journal of Veterinary Drug. 2016,50(09):41-46.

[2] Ma Xia, Guo Zhenhuan, Liu Yonglu, Wang Changlin, Zhang Guozu. Effects Comparision of Fuzheng Jiedu San and its Super Micropowder on Anti-IBDV in Vitro[J]. China Animal Husbandry & Veterinary Medicine. 2015,42(04):1032-1037.DOI:10.16431/j.cnki.1671-7236.2015.04.40.

[3] Dai Guangwen, Chen Sulian, Wei Bangwei, Jiang Jinghua, Xie Lihua, Wu Yinghang, Wu Zhiqiang. Screening test of different compound Chinese medicine preparations for prevention and treatment of swine hyperfebrile disease[J]. Chinese Journal of Traditional Veterinary Science. 2017(06):4-5.

[4] Li Z, Li L, Zhou H, Zeng L, Chen T, Chen Q, Zhou B, Wang Y, Chen Q, Hu P, Yang Z. Radix isatidis Polysaccharides Inhibit Influenza a Virus and Influenza A Virus-Induced Inflammation via Suppression of Host TLR3 Signaling In Vitro. Molecules. 2017,122(1):116-118. doi: 10.3390/molecules22010116. PMID: 28085062; PMCID: PMC6155848.

[5] Ding Y, Zeng L, Li R, Chen Q, Zhou B, Chen Q, Cheng PL, Yutao W, Zheng J, Yang Z, Zhang F. The Chinese prescription lianhuaqingwen capsule exerts anti-influenza activity through the inhibition of viral propagation and impacts immune function. BMC Complement Altern Med. 2017,17(1):130-142. doi: 10.1186/s12906-017-1585-7. PMID: 28235408; PMCID: PMC5324200.

4.The effects of FZJDS on uninfected cells were studied, but these effects were not analyzed. These effects are also necessary to analyze drug effects, so as to avoid non-specific concomitant effects.(3.6)

Ans: Thank you for your comment. In the present study, we analyzed the effects of FZJDS on uninfected cells in the mock group by western blot. Furthermore, we revised the discussion part hoping it better to elucidate the non-specific concomitant effects. We will be happy to edit the text further on the feedback comments on this study.

Reviewer #2: 

1.The antiviral effects of FZJDS were only tested in Marc-145 cell, which is originated from green monkey. The natural target cell of PRRSV is PAM, the authors should use PAM to perform assays to demonstrate the anti-PRRSV activity of FZJDS.

Ans: Thank you for your comment. In the present study, we selected sus scrofa species during the network pharmacologic analysis, besides previous studies have been widely elucidated that the mechanism of PRRSV infection on swine by the MARC‐145 cell line. In a study reported from Xun Wang et al found that tea polyphenol effectively inhibited PRRSV infection in Marc-145 cells by suppressing the stages of viral attachment, internalization, replication and release (Xun Wang et al. 2021). In line with previous publications showed that whole-genome transcriptome analysis of PRRSV-infected Marc-145 cells, which provides insight into the dynamics of host gene regulation during PRRSV infection (Ying Wei et al. 2018), as well as proved that PRRSV induced macropinocytosis via TIM-1 in MARC-145 cells (Xin Wei et al. 2020). Moreover, a proteomic study revealed that PRRSV virions incorporate a variety of simian proteins after replication in MARC-145 cells, which raises the intriguing possibility that cross-neutralization of PRRSV strains may, in part, be mediated by antibodies recognizing simian antigen (David Goldeck et al. 2019). Previous findings suggest that the role of the plasma membrane cholesterol in PRRSV infection of MARC-145 cells was investigated by Ying Sun et al (2011). While knockdown of DHX36 and its adaptor myeloid differentiation primary response gene 88 (MyD88) by small-interfering RNA in MARC-145 cells significantly reduced NF-kB activation and pro-inflammatory cytokine expression after PRRSV infection (Huiyuan Jing et al. 2017). Therefore, we used MARC‐145 cells in the present study for the cytological research validation.

We all totally agree with immune cells used to clarify the PRRSV infection mechanisms, rather than only either the MARC-145 cells or the immune effects. In the further research ,we'll both use immune cells for the mechanism. Thanks for your suggestions again.

[1] Wang X, Dong W, Zhang X, Zhu Z, Chen Y, Liu X, Guo C. Antiviral Mechanism of Tea Polyphenols against Porcine Reproductive and Respiratory Syndrome Virus. Pathogens. 2021 Feb 13;10(2):202. doi: 10.3390/pathogens10020202. PMID: 33668502; PMCID: PMC7917843.

[2] Wei Y, Li J, Zhang Y, Xue C, Cao Y. Tandem 3' UTR Patterns and Gene Expression Profiles of Marc-145 Cells During PRRSV Infection. Virol Sin. 2018 Aug;33(4):335-344. doi: 10.1007/s12250-018-0045-y. Epub 2018 Aug 1. PMID: 30069823; PMCID: PMC6178098.

[3] Wei X, Li R, Qiao S, Chen XX, Xing G, Zhang G. Porcine Reproductive and Respiratory Syndrome Virus Utilizes Viral Apoptotic Mimicry as an Alternative Pathway To Infect Host Cells. J Virol. 2020 Aug 17;94(17):e00709-20. doi: 10.1128/JVI.00709-20. PMID: 32522856; PMCID: PMC7431799.

[4] Goldeck D, Perry DM, Hayes JWP, Johnson LPM, Young JE, Roychoudhury P, McLuskey EL, Moffat K, Bakker AQ, Kwakkenbos MJ, Frossard JP, Rowland RRR, Murtaugh MP, Graham SP. Establishment of Systems to Enable Isolation of Porcine Monoclonal Antibodies Broadly Neutralizing the Porcine Reproductive and Respiratory Syndrome Virus. Front Immunol. 2019 Mar 27;10:572. doi: 10.3389/fimmu.2019.00572. PMID: 30972067; PMCID: PMC6445960.

[5] Sun Y, Xiao S, Wang D, Luo R, Li B, Chen H, Fang L. Cellular membrane cholesterol is required for porcine reproductive and respiratory syndrome virus entry and release in MARC-145 cells. Sci China Life Sci. 2011 Nov;54(11):1011-8. doi: 10.1007/s11427-011-4236-0. Epub 2011 Dec 16. PMID: 22173307; PMCID: PMC7088586.

[6] Jing H, Zhou Y, Fang L, Ding Z, Wang D, Ke W, Chen H, Xiao S. DExD/H-Box Helicase 36 Signaling via Myeloid Differentiation Primary Response Gene 88 Contributes to NF-κB Activation to Type 2 Porcine Reproductive and Respiratory Syndrome Virus Infection. Front Immunol. 2017 Oct 23;8:1365. doi: 10.3389/fimmu.2017.01365. PMID: 29123520; PMCID: PMC5662876.

2.The cytotoxicity of FZJDS on Marc-145 should be shown in graph rather than table. Further, the authors should calculate the CC50 of the drug.

Ans: Thank you for your suggestion. We calculate CC50 value and add some necessary information on cytotoxicity in this manuscript.

Supplementary Figure 1. Cytotoxicity of FZJDS on MARC-145 cells. (a) 24 hour FZJDS treatment group. (b) 36 hour FZJDS treatment group. (c) 48 hour FZJDS treatment group. (d) 72 hour FZJDS treatment group.

3.The authors also should calculate the IC50, which is indispensable for antiviral drug research.

Ans: Thank you for your suggestion. The IC50 value has been added.

4.In Figure 6C and E, the inhibitory effect of FZJDS on PRRSV replication is very weak, indicating that the drug may not be used for PRRSV control.

Ans: Thank you for your comment. We double check and analysis the results of N protein by Image J software, which is a marker response to PRRSV infection. When Marc-145 cells added without or with FZJDS as the negative and positive control group, compared with only PRRSV positive control group, the content of N protein was significantly increased.

5.The level of p-p65 in mock treatment is very high and is almost comparable to that of PRRSV infection groups. Commonly the p-p65 level in mock group is very low or cannot be detectable. This phenomenon is also observed in the level of p-JNK and p-IkBalpha. The data are questionable.

Ans: We apologize for the mistake of phosphorylated proteins band in mock group intensity analysis in the manuscript. We re-analyzed on this results carefully, hoping it is better for understanding. 

6.The data of bioinformatics shown in Figures 1-5 are too complex and difficult to follow, which makes the logic of the paper not clear.

Ans: Thank you for your comment. According to the analytical steps of network pharmacology, we list all the necessary results in line with the titles of each sections. Consistent with previous studies suggested that network pharmacology bioinformatics results in our study (Wenjun Zhou et al. 2021,Haoran Guo et al. 2021,Huahe Zhu et al. 2021). All the study demonstrate that network pharmacology is an effective tool for the discovery of natural compounds with specific properties and determination of their possible mechanisms. We will be happy to edit the text on the detailed comments on the network pharmacology results in this study.

[1] Zhou W, Zhu Z, Xiao X, Li C, Zhang L, Dang Y, Ge G, Ji G, Zhu M, Xu H. Jiangzhi Granule attenuates non-alcoholic steatohepatitis by suppressing TNF/NFκB signaling pathway-a study based on network pharmacology. Biomed Pharmacother. 2021 Nov;143:112181. doi: 10.1016/j.biopha.2021.112181. Epub 2021 Oct 4. PMID: 34649337.

[2] Guo H, Zeng H, Fu C, Huang J, Lu J, Hu Y, Zhou Y, Luo L, Zhang Y, Zhang L, Chen J, Zeng Q. Identification of Sitogluside as a Potential Skin-Pigmentation-Reducing Agent through Network Pharmacology. Oxid Med Cell Longev. 2021 Sep 23;2021:4883398. doi: 10.1155/2021/4883398. PMID: 34603597; PMCID: PMC8483913.

[3] Zhu H, Wang S, Shan C, Li X, Tan B, Chen Q, Yang Y, Yu H, Yang A. Mechanism of protective effect of xuan-bai-cheng-qi decoction on LPS-induced acute lung injury based on an integrated network pharmacology and RNA-sequencing approach. Respir Res. 2021 Jun 28;22(1):188. doi: 10.1186/s12931-021-01781-1. PMID: 34183011; PMCID: PMC8237774.

---

## [Decision Letter · Decision Letter 1]

21 Dec 2022

PONE-D-22-18907R1Fuzhengjiedu San inhibits porcine reproductive and respiratory syndrome virus by activating the PI3K/AKT pathwayPLOS ONE

Dear Dr. Chang,

Thank you for submitting your manuscript to PLOS ONE. After careful consideration, we feel that it has merit but does not fully meet PLOS ONE’s publication criteria as it currently stands. Therefore, we invite you to submit a revised version of the manuscript that addresses the points raised during the review process.

We look forward to receiving your revised manuscript.

Kind regards,

Ahmad Salimi

Academic Editor

PLOS ONE

Reviewers' comments:

Reviewer's Responses to Questions

**Comments to the Author**

1. If the authors have adequately addressed your comments raised in a previous round of review and you feel that this manuscript is now acceptable for publication, you may indicate that here to bypass the “Comments to the Author” section, enter your conflict of interest statement in the “Confidential to Editor” section, and submit your "Accept" recommendation.

Reviewer #1: All comments have been addressed

Reviewer #2: (No Response)

2. Is the manuscript technically sound, and do the data support the conclusions?

Reviewer #1: Partly

Reviewer #2: (No Response)

3. Has the statistical analysis been performed appropriately and rigorously? 

Reviewer #1: Yes

Reviewer #2: (No Response)

4. Have the authors made all data underlying the findings in their manuscript fully available?

Reviewer #1: Yes

Reviewer #2: (No Response)

5. Is the manuscript presented in an intelligible fashion and written in standard English?

Reviewer #1: Yes

Reviewer #2: (No Response)

6. Review Comments to the Author

Reviewer #1: In order to investigate the mechanism of FZJDS against Porcine reproductive and respiratory syndrome virus (PRRSV), this study used network pharmacology to predict the possible active ingredients and possible mechanisms of FZJDS. It was found that quercetin, kaempferol and luteolin in this drug may be the active ingredients. The (PI3K)/Akt pathway is considered to be a cellular target. For example, the expression levels of a variety of related proteins and genes were analyzed by molecular biology methods and compared with Internet drug studies. The results showed that FZJDS had significant antiviral activity by affecting the PI3K/AKT signaling pathway.

The (PI3K)/Akt pathway is a relatively good consideration to study the antiviral effect of traditional Chinese medicine in this study. However, there are some problems with the hypothesis and design that the authors should pay attention to.

1 The key evidence to study the antiviral effect of drugs is the influence of drugs on the proliferation level of the virus. After cells are infected with the virus, PCR or cell staining can be used for quantitative analysis of the virus, so as to preliminarily determine the antiviral effect of drugs. This study is only from the perspective of the cell effect of the virus, lacking the most critical antiviral effect. This does not meet the basic requirements of a complete scientific story.

2 The antiviral effect of cells should not be the primary function of the (PI3K)/Akt pathway, which is a key pathway for cell survival and, of course, one of the phenomena studied in this manuscript, the reduction of apoptosis effect. There are also more downstream molecules involved in the process of viral infection. For example, the mTOR gene can reduce the cell vitality of influenza virus infection, and the downregulation of its gene can enhance cell apoptosis. The downregulation of the gene expression can also reduce the virus titer in cells, indicating that mTOR can maintain the cell vitality of influenza virus infection and negatively regulate the apoptosis of influenza virus infection cells. The effective replication of influenza virus is ensured by maintaining cell viability and reducing cell apoptosis. The down-regulation of GSK-3β gene expression can reduce and increase the cell viability of influenza virus infection

The apoptosis induced by strong influenza virus decreased the titer of influenza virus in cells, suggesting that GSK-3β had a negative regulatory effect on apoptosis and promoted the replication of influenza virus by maintaining the vitality of host cells. All of these molecules are downstream of the (PI3K)/Akt pathway and are conducive to viral infection, but generally do not have antiviral effects. For example, studies have shown that Shufeng Xuanfei recipe and Jiebiao Qingli recipe can significantly down-regulate the mRNA transcription levels of AKT, Fas, FasL and Caspase-9 in the signaling pathway, and reduce the protein expression levels of AKT, Fas and FasL, thus playing an anti-influenza role.

3 There are two types of cellular antiviral effects. One is the production of molecules that kill the virus, such as antibodies and interferon. Instead, the cell commits suicide, allowing the infected virus to disappear along with the cell. From this perspective, the signaling pathways that promote cell survival are not conducive to the antiviral purposes of multicellular organisms.

Has a network pharmacological approach been used to study the antiviral molecular basis of this drug, but the effect of this drug on the (PI3K)/Akt pathway has not been found? How does the author explain why similar research methods and objectives produce such different conclusions?DOI:10.16466/j.issn1005-5509.2020.08.012.

Reviewer #2: (No Response)

7. PLOS authors have the option to publish the peer review history of their article (what does this mean?). If published, this will include your full peer review and any attached files.

Reviewer #1: No

Reviewer #2: No

---

## [Author Response · Author response to Decision Letter 1]

8 Feb 2023

Dear editor and reviewer,

We greatly appreciate the reviewer’s detailed and valuable comments on our manuscript entitled “Fuzhengjiedu San inhibits porcine reproductive and respiratory syndrome virus by activating the PI3K/AKT pathway” (ID: PONE-D-22-18907). On behalf of my coauthors, I would like to clarify our responses below to all the points raised by the reviewers. We hope that the reviewers and the editor will be satisfied with the changes and additions made to the original manuscript in the revised file. Their critiques have made the paper better and we thank you again for your kind consideration.

The responses to the reviewer’s comments and details of the corrections follow:

Reviewer #1

1. The key evidence to study the antiviral effect of drugs is the influence of drugs on the proliferation level of the virus. After cells are infected with the virus, PCR or cell staining can be used for quantitative analysis of the virus, so as to preliminarily determine the antiviral effect of drugs. This study is only from the perspective of the cell effect of the virus, lacking the most critical antiviral effect. This does not meet the basic requirements of a complete scientific story.

Ans: Thank you for your comment. We strongly agree with your suggestion about quantitating viral proliferation to make a more complete story and this is planned for the follow-up study. In the present study, we followed the pattern of previously published work and used RT-qPCR and western blotting to determine the inhibitory effects of FZJDS on PRRSV, in line with other researchers’ methods for testing the antiviral effect. For example, Ruansit, et al. [1] evaluated the in vitro antiviral activities and the ex vivo immunomodulatory effects of Houttuynia cordata Thunb through RT-qPCR, and the in vitro MTT assay was used by Jiang, et al. [2] to measure the effect of Isatis root polysaccharide on PRRSV infection. To assess the inhibitory activity of another key component of FZJDS, RT-qPCR was employed by Su Ke et al. [3] to determine the effect of radix isatidis on the replication pathway and proliferation of PRRSV. Our current study followed previous experimental methods, but we do plan to perform cell staining in a subsequent analysis of the antiviral effect and determine actual viral titers. Thanks again for this valuable suggestion.

References:

[1] Ruansit W, Charerntantanakul W. Oral Supplementation of Houttuynia cordata Extract Reduces Viremia in PRRSV-1 Modified-Live Virus-Vaccinated Pigs in Response to the HP-PRRSV-2 Challenge. Front Immunol. 2022 Jul 18;13:929338. doi: 10.3389/fimmu.2022.929338.

[2] Jiang D, Zhang L, Zhu G, Zhang P, Wu X, Yao X, Luo Y, Yang Z, Ren M, Wang X, Chen S, Wang Y. The Antiviral Effect of Isatis Root Polysaccharide against NADC30-like PRRSV by Transcriptome and Proteome Analysis. Int J Mol Sci. 2022 Mar 28;23(7):3688. doi: 10.3390/ijms23073688.

[3] DOI: 10.16303/j.cnki.1005-4545.2022.12.24.

2.The antiviral effect of cells should not be the primary function of the (PI3K)/Akt pathway, which is a key pathway for cell survival and, of course, one of the phenomena studied in this manuscript, the reduction of apoptosis effect. There are also more downstream molecules involved in the process of viral infection. For example, the mTOR gene can reduce the cell vitality of influenza virus infection, and the downregulation of its gene can enhance cell apoptosis. The downregulation of the gene expression can also reduce the virus titer in cells, indicating that mTOR can maintain the cell vitality of influenza virus infection and negatively regulate the apoptosis of influenza virus infection cells. The effective replication of influenza virus is ensured by maintaining cell viability and reducing cell apoptosis. The down-regulation of GSK-3β gene expression can reduce and increase the cell viability of influenza virus infection.

The apoptosis induced by strong influenza virus decreased the titer of influenza virus in cells, suggesting that GSK-3β had a negative regulatory effect on apoptosis and promoted the replication of influenza virus by maintaining the vitality of host cells. All of these molecules are downstream of the (PI3K)/Akt pathway and are conducive to viral infection, but generally do not have antiviral effects. For example, studies have shown that Shufeng Xuanfei recipe and Jiebiao Qingli recipe can significantly down-regulate the mRNA transcription levels of AKT, Fas, FasL and Caspase-9 in the signaling pathway, and reduce the protein expression levels of AKT, Fas and FasL, thus playing an anti-influenza role.

Ans: Thank you for taking the time to make this detailed comment. Your insights and information were very helpful for revising and improving our paper. In this study, the key reason why we selected the PI3K/AKT pathway is that it was the predominant result indicated by KEGG enrichment analysis as well as involved the core proteins most associated with PRRSV infection. Based on references from pharmacological network searching, the PI3K/AKT pathway was most often selected to verify antiviral activity by western immunoblotting and flow cytometry. A recent genome-wide mRNA and long non-coding RNA (lncRNA) search and analysis in PRRSV-infected PTR2 showed that PRRSV infection could inhibit the PI3K/AKT/mTOR pathway and trigger apoptosis [1]. Another study determined the phosphorylation status of Akt during infection of MARC-145 cells with the highly pathogenic PRRSV strain, HuN4, and provided new evidence of a novel role for the PI3K/Akt pathway in PRRSV infection [2]. Thus, we have modified the discussion section to include Bax, Bcl-2, TLR4 and p53, which have been identified as the core target proteins of FZJDS, in line with a previous report that protocatechuic acid suppressed H1N1-induced TLR4/NF-κB activation [3]. Previous studies revealed that bovine ephemeral fever virus disrupted Beclin 1 and Bcl-2 interaction by JNK-mediated Bcl-2 phosphorylation, thereby activating autophagy [4], which proves that JNK and Bcl-2 are among the chief proteins affecting PRRSV proliferation. Activated JNK translocates to the mitochondria where it phosphorylates Bcl-2/Bcl-xL and antagonizes the anti-apoptotic activity of Bcl-2/Bcl-xL. We incorporated these results into the present study discussion in hopes of clarifying the antiviral mechanism according to your comments and making the paper better. We would be happy to take any further feedback.

References:

[1] Zhang X, Liu X, Peng J, Song S, Xu G, Yang N, Wu S, Wang L, Wang S, Zhang L, Liu Y, Liang P, Hong L, Xu Z, Song C. Genome-Wide mRNA and Long Non-Coding RNA Analysis of Porcine Trophoblast Cells Infected with Porcine Reproductive and Respiratory Syndrome Virus Associated with Reproductive Failure. Int J Mol Sci. 2023 Jan 4;24(2):919. doi: 10.3390/ijms24020919.

[2] Zhu L, Yang S, Tong W, Zhu J, Yu H, Zhou Y, Morrison RB, Tong G. Control of the PI3K/Akt pathway by porcine reproductive and respiratory syndrome virus. Arch Virol. 2013 Jun;158(6):1227-34. doi: 10.1007/s00705-013-1620-z.

[3] Qian Wang, Xiaojuan Ren, Jinhua Wu, Hongrong Li, Liu Yang, Yan Zhang, Xin Wang, Zhicun Li, Eur J Clin Microbiol Infect Dis. 2022 Jan 24;1-8. https://doi.org/10.1007/s10096- 022-04401-y.

[4] Ching-Yuan Cheng, Hsu-Hung Tseng, Hung-Chuan Chiu, Ching-Dong Chang, Brent L. Nielsen, Vet Res (2019) 50:79, https://doi.org/10.1186/s13567-019-0697-0.

3. There are two types of cellular antiviral effects. One is the production of molecules that kill the virus, such as antibodies and interferon. Instead, the cell commits suicide, allowing the infected virus to disappear along with the cell. From this perspective, the signaling pathways that promote cell survival are not conducive to the antiviral purposes of multicellular organisms.

Has a network pharmacological approach been used to study the antiviral molecular basis of this drug, but the effect of this drug on the (PI3K)/Akt pathway has not been found? How does the author explain why similar research methods and objectives produce such different conclusions?

Ans: We greatly appreciate your insightful comments and suggestions. We have read the article (DOI:10.16466/j.issn1005-5509.2020.08.012.) and found that the main components of the Fuzheng Jiedu formulation used in that study were white Rhizoma atrictyloides, tangerine peel, parsnip, licorice, patchouli, Astragalus membranaceus, honeysuckle and forsythiae. However, the Fuzheng Jiedu San used by us is a combination of Isatis root, Astragalus membranaceus and Herba medii. Due to significant differences in drug compatibility as delineated by network pharmacological analysis, the components, targets and pathways of these formulations are totally different. 

The drug compatibility of traditional Chinese medicine preparations may increase or decrease according to the disease, and there has been no network pharmacological antiviral study on a formulation with the same components as used in this study; however, there have been reports on related single drugs such as from Abelmoschus manihot that can inhibit IAV-induced lung inflammation [1]. A network pharmacology investigation of the anti-inflammatory and antiviral effects of Isatis indigotica identified lignans that might regulate the immune system by targeting IL-2 and the PI3K/Akt signaling pathway [2]. Radix isatidis was analyzed through network pharmacology technology and it was found that COVID-19 could be inhibited by targeting the signaling pathways involving receptors for advanced glycation end products, interleukin-17, tumor necrosis factor, sphingolipids, and p53 [3].

References:

[1] Gao Y, Liang Z, Lv N, Shan J, Zhou H, Zhang J, Shi L. Exploring the total flavones of Abelmoschus manihot against IAV-induced lung inflammation by network pharmacology. BMC Complement Med Ther. 2022 Feb 5;22(1):36. doi: 10.1186/s12906-022-03509-0.

[2] Deng J, Ma Y, He Y, Yang H, Chen Y, Wang L, Huang D, Qiu S, Tao X, Chen W. A Network Pharmacology-Based Investigation to the Pharmacodynamic Material Basis and Mechanisms of the Anti-Inflammatory and Anti-Viral Effect of Isatis indigotica. Drug Des Devel Ther. 2021 Jul 20;15:3193-3206. doi: 10.2147/DDDT.S316701.

[3] Yu B, Lin F, Ning H, Ling B. Network pharmacology study on the mechanism of the Chinese medicine Radix Isatidis (Banlangen) for COVID-19. Medicine (Baltimore). 2021 Aug 13;100(32):e26881. doi: 10.1097/MD.0000000000026881.

We appreciate the reviewer’s detailed and insightful comments and hope that the revised manuscript will meet with their approval. Thanks to all for your help.

Sincerely,

Jinjin Tong

---

## [Decision Letter · Decision Letter 2]

15 Mar 2023

Fuzhengjiedu San inhibits porcine reproductive and respiratory syndrome virus by activating the PI3K/AKT pathway

PONE-D-22-18907R2

Dear Dr.Kexin Chang ,

We’re pleased to inform you that your manuscript has been judged scientifically suitable for publication and will be formally accepted for publication once it meets all outstanding technical requirements.

Kind regards,

Ahmad Salimi

Academic Editor

PLOS ONE

Reviewers' comments:

Reviewer's Responses to Questions

**Comments to the Author**

1. If the authors have adequately addressed your comments raised in a previous round of review and you feel that this manuscript is now acceptable for publication, you may indicate that here to bypass the “Comments to the Author” section, enter your conflict of interest statement in the “Confidential to Editor” section, and submit your "Accept" recommendation.

Reviewer #1: All comments have been addressed

Reviewer #2: (No Response)

2. Is the manuscript technically sound, and do the data support the conclusions?

Reviewer #1: Yes

Reviewer #2: (No Response)

3. Has the statistical analysis been performed appropriately and rigorously? 

Reviewer #1: I Don't Know

Reviewer #2: (No Response)

4. Have the authors made all data underlying the findings in their manuscript fully available?

Reviewer #1: Yes

Reviewer #2: (No Response)

5. Is the manuscript presented in an intelligible fashion and written in standard English?

Reviewer #1: Yes

Reviewer #2: (No Response)

6. Review Comments to the Author

Reviewer #1: The general principle of the treatment of viral infection in pigs should also consider the economic factors, for some malignant infectious virus infection, currently to eliminate the source of infection, environmental disinfection and blocking infection as the basic strategy. From a medical perspective, the search for and exploration of treatable tools still deserves encouragement. This study used TCM compound to study the therapeutic effect of porcine virus infection. Has some value from a veterinary perspective. This study can also provide relevant evidence for the treatment of human virus infection. From the perspective of human clinical order, it also has some significance. However, viral infectious diseases should be carried out around the core issue of viral infection. The present manuscript study is slightly inadequate in this regard.

Reviewer #2: (No Response)

7. PLOS authors have the option to publish the peer review history of their article (what does this mean?). If published, this will include your full peer review and any attached files.

Reviewer #1: No

Reviewer #2: No

---

## [Editor Report · Acceptance letter]

12 Apr 2023

PONE-D-22-18907R2 

Fuzhengjiedu San inhibits porcine reproductive and respiratory syndrome virus by activating the PI3K/AKT pathway 

Dear Dr. Chang:

I'm pleased to inform you that your manuscript has been deemed suitable for publication in PLOS ONE. Congratulations! Your manuscript is now with our production department. 

Kind regards, 

on behalf of

Dr. Ahmad Salimi 

Academic Editor

PLOS ONE